# Data Mixing Can Induce Phase Transitions in Knowledge Acquisition

**Xinran Gu[1,3]***  **Kaifeng Lyu[1]*†**  **Jiazheng Li[2,3]**  **Jingzhao Zhang[1,3‡]**

[1]Institute for Interdisciplinary Information Sciences, Tsinghua University
[2]College of AI, Tsinghua University    [3]Shanghai Qizhi Institute
`guxr24@mails.tsinghua.edu.cn, klyu@tsinghua.edu.cn`
`foreverlasting1202@outlook.com, jingzhaoz@tsinghua.edu.cn`

## Abstract

Large Language Models (LLMs) are typically trained on data mixtures: most data come from web scrapes, while a small portion is curated from high-quality sources with dense domain-specific knowledge. In this paper, we show that when training LLMs on such data mixtures, knowledge acquisition from knowledge-dense datasets—unlike training exclusively on knowledge-dense data [Allen-Zhu and Li, 2024a]—*does not* always follow a smooth scaling law but can exhibit phase transitions with respect to the mixing ratio and model size. Through controlled experiments on a synthetic biography dataset mixed with web-scraped data, we demonstrate that: (1) as we increase the model size to a critical value, the model suddenly transitions from memorizing very few to most of the biographies; (2) below a critical mixing ratio, the model memorizes almost nothing even with extensive training, but beyond this threshold, it rapidly memorizes more biographies. We attribute these phase transitions to a capacity allocation phenomenon: a model with bounded capacity must act like a knapsack problem solver to minimize the overall test loss, and the optimal allocation across datasets can change discontinuously as the model size or mixing ratio varies. We formalize this intuition in an information-theoretic framework and reveal that these phase transitions are predictable, with the critical mixing ratio following a power-law relationship with the model size. Our findings highlight a concrete case where a good mixing recipe for large models may not be optimal for small models, and vice versa.

## 1 Introduction

The pre-training data of large language models (LLMs) can be categorized into two major types. The first type consists of large-scale corpora scraped from the web [Raffel et al., 2020, Penedo et al., 2024, Li et al., 2024], often spanning billions to trillions of tokens across diverse topics and styles. Due to the scale, it is inherently hard to ensure the information density of the dataset and its relevance to downstream tasks. Hence, a second type of data, smaller-scale datasets curated from high-quality sources, is incorporated. This type of data usually contains very dense knowledge on tasks or domains with significant practical value. For example, Wikipedia and Stack Exchange cover a wide range of world knowledge. OpenWebMath [Paster et al., 2024] and StarCoder [Li et al., 2023, Kocetkov et al., 2022] provide valuable data for improving model performance on mathematics and coding tasks.

The second type of data, which we refer to as knowledge-dense data, typically accounts for only a small fraction of the entire corpus. In the pre-training data of a recently released model family,

---

*Equal contribution
†Work done while at the Simons Institute for the Theory of Computing, UC Berkeley.
‡Corresponding author

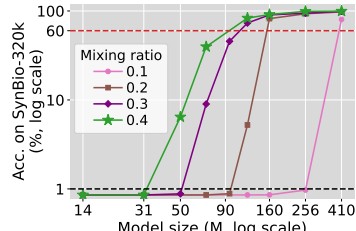
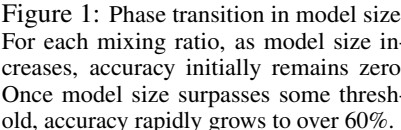

Figure 1: Phase transition in model size. For each mixing ratio, as model size increases, accuracy initially remains zero. Once model size surpasses some threshold, accuracy rapidly grows to over 60%.

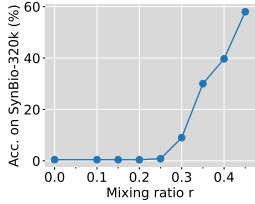
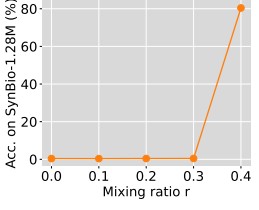

(a) 70M models.      (b) 410M models.

Figure 2: Phase transition in mixing ratio. For each model size, as mixing ratio $r$ increases, accuracy initially remains zero. Only when $r$ exceeds some threshold does accuracy quickly improve.

OLMo 2 [OLMo et al., 2025], over 95% of the tokens are from web data, and only less than 5% are from knowledge-dense data. The proportion of each individual knowledge-dense dataset is even smaller, *e.g.*, only less than 0.1% of the tokens are from Wikipedia. This naturally raises a question: *How much knowledge can LLMs really acquire from this small amount of knowledge-dense data?*

If LLMs were *exclusively* trained on knowledge-dense data without any data mixing, the amount of knowledge acquired after sufficient training should scale linearly with *model size*. Although quantifying knowledge in natural data is non-trivial, Allen-Zhu and Li [2024a] sidestep this issue and provide strong empirical evidence for this linear scaling law through extensive pre-training experiments on synthetically generated biographies. In their setting, the amount of knowledge stored by a model is quantified by evaluating its memorization of the biographies using information-theoretic metrics. Similar linear scaling laws are also observed in memorizing Wikidata fact triples by Lu et al. [2024], and analyzed theoretically by Nichani et al. [2025]. Based on these results, one might naively expect a similar linear relationship between model size and acquired knowledge when knowledge-dense data is mixed with web data.

However, in this paper, we show that *the linear scaling no longer holds under data mixing*. We consider the setup where a knowledge-dense dataset focused on a single domain constitutes a small fraction $r$ of the pre-training corpus—referred to as the *mixing ratio*—and the rest is large-scale web text (see Appendix E.1 for our implementation of data mixing). We demonstrate via a quantitative study that knowledge acquisition from the knowledge-dense data exhibits a more intricate behavior with notable phase transitions with respect to the mixing ratio and model size.

More specifically, we study factual knowledge acquisition. We follow the approach of Allen-Zhu and Li [2024a] to curate a synthetic dataset of biographies, where each individual's information is embedded into natural text descriptions using diverse templates. Due to the uniform data format and content of this dataset, we can quantify how much knowledge the model has stored simply by counting the number of memorized biographies. We then mix this synthetic biography dataset with large-scale web corpus FineWeb-Edu [Penedo et al., 2024] or the Pile [Gao et al., 2020] to create the pre-training mixture. We pre-train or continually pre-train Pythia models [Biderman et al., 2023] ranging from 14M to 6.9B parameters on these mixtures.

While setting $r$ closer to 1 will make the model learn more from the knowledge-dense data, in practice, $r$ is typically set to a small value either because the knowledge-dense data has limited amount or increasing $r$ may hurt the model's capabilities acquired from other domains. Therefore, the essence of our study is to understand whether models can still memorize a decent number of biographies for relatively small $r$. Our experiments reveal two interesting findings (Section 3):

**Finding 1: Phase Transition in Model Size (Figure 1).** Fixing the mixing ratio $r$ and varying the model size $M$, we observe that when $M$ is smaller than a critical model size $M_{\text{thres}}$, the number of memorized biographies can be nearly zero. Only when $M > M_{\text{thres}}$, the model *suddenly* memorizes most biographies. Moreover, the threshold $M_{\text{thres}}$ is higher for smaller $r$.

**Finding 2: Phase Transition in Mixing Ratio (Figures 2 and 9).** When varying the mixing ratio $r$ while keeping the model size $M$ fixed, we find that below a critical mixing ratio $r_{\text{thres}}$, the model memorizes almost nothing even after significantly longer training, during which each biography appears hundreds of times or more (Figures 3(a) and 4). But when $r > r_{\text{thres}}$, the number of memorized biographies grows rapidly with $r$. We further find that as we gradually decrease $r$, the number of steps needed to memorize a fixed number of biographies initially grows linearly with

$1/r$ (Figure 3(b)), but soon becomes exponential and even superexponential (Figure 3(c)), making it impossible or practically infeasible for the model to memorize a non-trivial number of biographies.

In Figure 10, we further show that the observed phase transitions are not limited to discrete metrics like accuracy, but also persist in validation loss, a continuous metric.

**Theoretical Analysis.** In Section 4, we attribute the observed phase transitions to a capacity allocation phenomenon: a model with bounded capacity must act like a knapsack problem solver to minimize the overall test loss, and the optimal allocation across datasets can change discontinuously as the model size or mixing ratio varies. To formalize this intuition, we model a sufficiently trained LLM as the best model that minimizes the test loss under a fixed capacity constraint $M$. We develop an information-theoretic framework and show that, when trained on a mixture of knowledge-dense and web-scraped data, the model should allocate its capacity across the two datasets based on their respective "marginal values"—that is, the reduction in test loss achieved by assigning one additional unit of capacity to that dataset. We rigorously prove that only when the mixing ratio $r$ or the model size $M$ is above a certain threshold does the knowledge-dense dataset become worth learning, thus leading to the observed phase transitions. Assuming that the optimal test loss on web-scraped data follows a power law in model size, we further show that these phase transitions are in fact predictable, with the critical mixing ratio following a power-law relationship with the model size. Empirically, we validate this power-law relationship on both synthetic biographies and a set of real-world knowledge extracted from Wikipedia (Section 5).

**Strategies to Enhance Knowledge Acquisition Under Low Mixing Ratios (Section 6).** Inspired by our theory, we propose two strategies to enhance knowledge acquisition at low mixing ratios: (1) randomly subsampling the knowledge-dense dataset; (2) rephrasing knowledge into more compact forms and augmenting the original dataset with the rephrased versions. The key idea is to increase the "marginal value" of the knowledge-dense dataset by increasing the exposure frequency of each single fact. We validate on both synthetic and real-world Wikipedia biographies that these strategies help models memorize significantly more biographies while preserving models' general capability.

**Takeaways.** The key takeaways of our paper are as follows:

1. The mixing ratio should be set with care for different model sizes: mixing in knowledge-dense datasets with small mixing ratios can offer no benefit at all, especially when training small LMs.

2. Naively measuring the performance of small models on a small data domain may provide little to no predictive signal on how well larger models perform, revealing a potential limitation of using small proxy models for data curation, as also evidenced by Kang et al. [2024], Jiang et al. [2024], Ye et al. [2024], Magnusson et al. [2025], Mizrahi et al. [2025].

3. Slightly improving the "marginal value" of knowledge-dense data can offer a large gain in performance, as evidenced by our proposed strategies.

## 2 Experimental Setup

**The SynBio Dataset.** We follow Allen-Zhu and Li [2024b] to create a synthetic biography dataset, with each individual characterized by five attributes: birth date, birth city, university, major, and employer. For each individual, the value of each attribute is randomly and independently sampled from a predefined domain. These (name, attribute, value) triplets are then converted into natural text using sentence templates. For instance, (Gracie Tessa Howell, birth city, St. Louis, MO) is converted into "Gracie Tessa Howell's birthplace is St. Louis, MO." Following [Allen-Zhu and Li, 2024b], every time the model encounters a biography, the five sentences are randomly shuffled, and a new sentence template is selected for each attribute from a set of five possible templates. We denote the dataset containing $N$ biographies as SynBio-$N$. See Appendix E.2.1 for full details.

**Evaluation.** Denote a knowledge triplet (name, attribute, value) as $(\boldsymbol{n}, \boldsymbol{a}, \boldsymbol{v})$ and let $|\boldsymbol{v}|$ represent the number of tokens in $\boldsymbol{v}$. For evaluation, the model is prompted with the sentence prefix containing $\boldsymbol{n}$ and $\boldsymbol{a}$ and is tasked to generate $|\boldsymbol{v}|$ tokens via greedy decoding. We then check whether the output exactly matches $\boldsymbol{v}$. For example, given the triplet (Gracie Tessa Howell, birth city, St. Louis, MO), the prompt "Gracie Tessa Howell's birthplace is" is provided. We say the model has memorized the fact if it generates "St. Louis, MO." We report the accuracy averaged over all individuals, attributes, and templates in the main text and defer the detailed results to Appendix D.6.

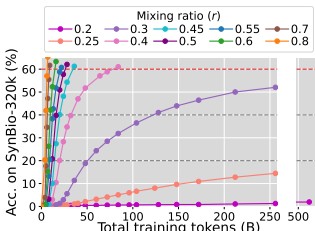 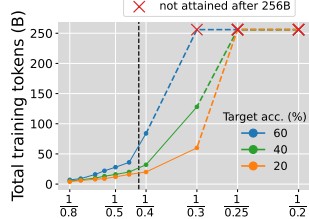 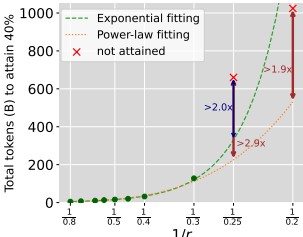

(a) Train until acc. 60% or a total of 256B tokens are passed.

(b) Required training steps to achieve target accuracy v.s $1/r$.

(c) Fitting required training steps to attain 40% accuracy against $1/r$.

Figure 3: Training longer barely helps for low mixing ratios, with the required training steps to reach a target accuracy grow exponentially or even superexponentially with $1/r$. We train 70M models on the mixture of FineWeb-Edu and SynBio-320k with $r$ ranging from 0.2 to 0.8.

**Training Setup.** Our experiments use the Pythia architecture [Biderman et al., 2023], with model sizes ranging from 14M to 6.9B. The default setup involves pre-training from scratch on a mixture of FineWeb-Edu and SynBio. Since FineWeb-Edu is large (>1T tokens) and SynBio is small (<1B tokens), our typical training runs involve the model seeing SynBio for multiple epochs but FineWeb-Edu for less than one epoch. For instance, in a 32B-token run with the mixing ratio for SynBio-320k set as 0.1, the model passes SynBio $\sim 100$ times. We also study the continual pre-training setup in Section 6 and Appendix D.1. Full experimental details are provided in Appendix E.

## 3 Phase Transitions of Knowledge Acquisition within Data Mixtures

### 3.1 Phase Transition in Model Size

We first investigate how knowledge acquisition is affected by model size given the data mixture. For each $r \in \{0.1, 0.2, 0.3, 0.4\}$, we train models with sizes from 14M to 410M on the mixture of FineWeb-Edu and SynBio-320k for a sufficiently long horizon of 32B tokens, which is approximately four times the compute-optimal training tokens for 410M models according to Hoffmann et al. [2022]. As shown in Figure 1, as the model size increases, accuracy on SynBio initially remains near zero. Once the model size surpasses some threshold, accuracy rapidly grows to above 60%. The transition is consistently sharp across different mixing ratios while larger $r$ leads to a smaller critical point.

### 3.2 Phase Transition in Mixing Ratio

We now study how knowledge acquisition under data mixing scenario is affected by mixing ratios.

**Performance on knowledge-dense data undergoes a phase transition as mixing ratio increases.** We begin by training models of the same size with different mixing ratios $r$. Specifically, we train 70M models on the mixture of FineWeb-Edu and SynBio-320K, varying $r$ from 0.1 to 0.45 (stepsize 0.05), and 410M models on the mixture of FineWeb-Edu and SynBio-1.28M, varying $r$ from 0.1 to 0.4 (stepsize 0.1). All models are trained for a total of 32B tokens. As shown in Figure 2(a), for 70M models, as $r$ increases from 0.1 to 0.25, its accuracy on SynBio remains near zero. Only when $r > 0.3$ does the accuracy begin to steadily improve. In Figure 2(b), the accuracy for 410M models exhibit similar trends where it remains near zero for $r \leq 0.3$ and suddenly attains 80% when $r$ grows to 0.4. In Figure 9, we replicate the experiments on Pythia 2.8B and 6.9B models to show that similar phase transition in mixing ratio persists for larger models. In Table 1, we report the mean and standard deviation of accuracy for experiments in Figure 2(a).

**Training longer barely helps for low mixing ratios.** Given the observed phase transition, one may raise the following counter-argument: if models are trained for a sufficiently long horizon—such that even a small mixing ratio $r$ would eventually result in each biography being encountered hundreds or even thousands of times—then the phase transition might no longer exist. To test this counter-argument, we extend the training horizon for $r = 0.2$ to 512B tokens for the 70M and 410M models by 16 and 4 times respectively. Under this extended training, each biography appears $\sim 3000$ times for the 70M model and $\sim 200$ times for the 410M model. As shown in Figures 3(a) and 4, the accuracy on SynBio remains near zero even after such extensions.

**Required training steps increase exponentially or even superexponentially with $1/r$.** To further refute this counter-argument, we quantify how the required training steps to reach a target accuracy, denoted as $T$, scales with $1/r$. Specifically, we train 70M models with $r$ ranging from 0.2 to 0.8. For

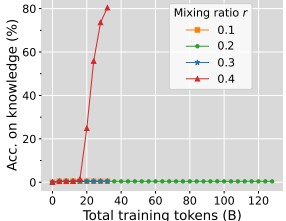
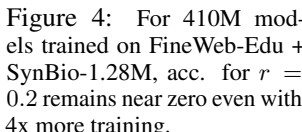

Figure 4: For 410M models trained on FineWeb-Edu + SynBio-1.28M, acc. for $r = 0.2$ remains near zero even with 4x more training.

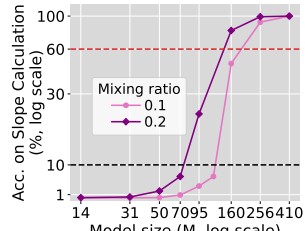
(a) Phase transition in model size.

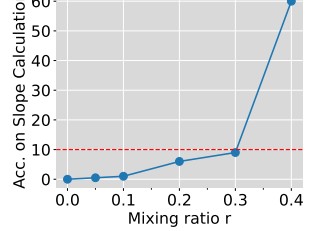
(b) Phase transition in $r$.

Figure 5: Similar phase transitions for the slope calculation subtask persist when we mix the modified OpenWebMath with FineWeb-Edu. The model size for (b) is 70M.

each mixing $r$, we evaluate 20 training horizons, approximately evenly spaced on a logarithmic scale with a factor of 1.2 ranging from 0 to 256B tokens. Training continues until the model reaches 60% accuracy or exhausts 256B tokens. As shown in Figures 3(a) and 3(b), when $r$ decreases from 0.8, $T$ initially increase linearly with $1/r$ for $r > 0.4$ and quickly deviates from the linear trend for $r < 0.4$.

We further fit a scaling law the required training steps to reach 40% accuracy against $1/r$, modeling $T$ as a power-law or exponential function of $1/r$. Specifically, we fit $T$ against $1/r$ for $r \geq 0.3$ and examine whether the extrapolation can predict $T$ for smaller $r$. As shown in In Figure 3(c), the actual $T$ is more than 2.9 times the power-law prediction for $r = 0.25$, and more than 1.9 times for $r = 0.2$. Moreover, the actual $T$ for $r = 0.25$ is even more than twice the exponential prediction. These significant deviations suggest exponential or even superexponential growth of $T$ with respect to $1/r$. See Appendix E.5 for the detailed fitting process.

We also conduct ablation studies on hyperparameters in Appendix D.2.

### 3.3 Phase Transitions on Reasoning Tasks

In this subsection, we show that the phase transition phenomenon is not limited to factual knowledge, but also extends to datasets related to reasoning. Such datasets are often multi-task in practice. For example, OpenWebMath [Paster et al., 2024] covers diverse math topics. We show that phase transitions can occur for each single subtask within this dataset. Inspired by Ruis et al. [2024], we consider the slope calculation task between two points $(x_1, y_1)$ and $(x_2, y_2)$. To explicitly control the frequency and format of the slope calculation examples, we replace all the documents containing the word "slope" in OpenWebMath with our clean and high-quality slope calculation demonstrations. We then mix the modified OpenWebMath with FineWeb-Edu and train Pythia models on this mixture from scratch. Similar to the setup of SynBio, every time the model sees a slope calculation example, we uniformly sample $x_1, y_1, x_2, y_2$ from $\{0, 1, \cdots, 99\}$ (ensuring $x_1 \neq x_2$), and apply randomly chosen question-answer templates. For evaluation, we randomly generate 1k questions for slope calculation and check if the model produces the correct final answer. Results in Figure 5 show similar phase transitions as factual knowledge acquisition (see details in Appendix E.3). Appendix D.3 presents further discussions and experiments on another reasoning task with larger input space.

## 4 Theoretical Analysis

In this section, we take an information-theoretic view to explain the observed phase transitions. The key challenge in developing a theory is that training LLMs can involve a lot of tricks, making it hard to identify the most important factors in inducing the phase transitions. In our paper, we consider an ideal case where the model is sufficiently trained, allowing us to focus on the key factor—model capacity—and abstract away all other complexities.

### 4.1 High-Level Intuition

We model a sufficiently trained language model with capacity $M$ as an *optimal bounded-capacity learner*, which minimizes test loss as much as possible under the capacity constraint $M$. The high-level intuition can be framed as a fractional knapsack problem (see Figure 6 for an illustration).

When training solely on knowledge-dense data, where each fact appears with equal probability, the optimal learner seeks to store as much knowledge as possible within its capacity. As a result, the total amount of memorized knowledge scales proportionally with the model's capacity $M$ (Section 4.3).

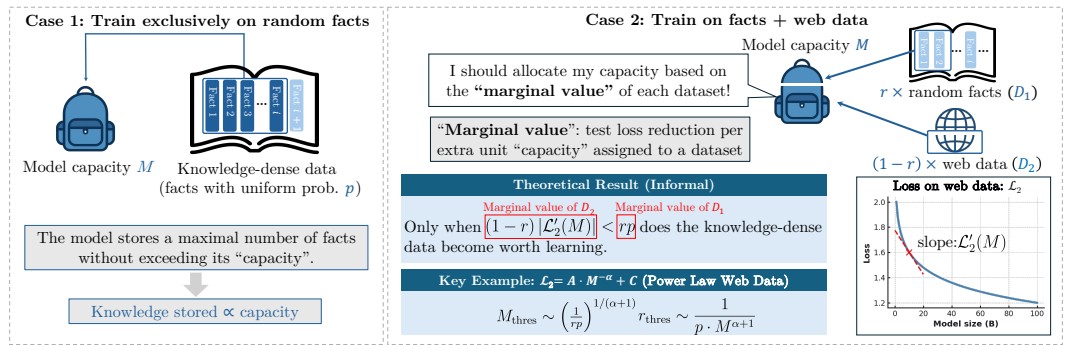

Figure 6: An illustration of the intuition behind our theory.

However, the situation changes when the knowledge-dense data is mixed with web-scraped data. In this case, the optimal learner should allocate its capacity across the two datasets based on their respective "marginal values"—that is, the reduction in test loss resulting from assigning one additional unit of capacity to a dataset. Only when $r$ or $M$ exceeds a certain threshold does the knowledge-dense data become worth learning.

## 4.2 Problem Formulation

**Data distribution.** The essence of language modeling is to model the distribution of the next token $y$ for a given context $x$ containing all previous tokens. We take a Bayesian view, assuming a latent variable $\theta \in \Theta$ governing the distribution of $(x, y)$, denoted as $(x, y) \sim \mathcal{D}_\theta$. Conceptually, $\theta$ encodes knowledge about the world. For example, a person may be born in 1996 in one universe but 1999 in another. Or, in a different universe, popular Python libraries may feature a different set of functions. We assume the universe first draws $\theta$ from a prior $\mathcal{P}$ before we observe the data distribution $\mathcal{D}_\theta$.

**Learning Algorithm.** A learning algorithm $\mathcal{A}$ is a procedure that takes samples from a data distribution $\mathcal{D}$ of $(x, y)$ and outputs a predictor $h = \mathcal{A}(\mathcal{D})$, which maps $x$ to a distribution over $y$. The performance of $h$ is measured by the expected cross-entropy loss $\mathcal{L}(h; \mathcal{D}) := \mathbb{E}_{(x,y)\sim\mathcal{D}}[-\log p(y \mid h, x)]$, where $p(y \mid h, x)$ denotes the predicted distribution of $y$ given $x$ by the predictor $h$, and $\log$ is in base 2 for convenience. We measure the performance of a learning algorithm $\mathcal{A}$ by its expected loss over all data distributions $\mathcal{D}_\theta$ with respect to the prior $\mathcal{P}$:

$$\bar{\mathcal{L}}_\mathcal{P}(\mathcal{A}) := \mathbb{E}_{\theta\sim\mathcal{P}}[\mathcal{L}(\mathcal{A}(\mathcal{D}_\theta); \mathcal{D}_\theta)]. \tag{1}$$

In practice, a predictor $h$ can be a transformer, and $\mathcal{A}$ can be the pre-training algorithm.

**Model Capacity and Mutual Information.** We measure a model's "effective" capacity—the amount of information a model, produced by some learning algorithm $\mathcal{A}$, stores about the data distribution $\mathcal{D}_\theta$—by the mutual information (MI) between the model and the data distribution $\mathcal{D}_\theta$, i.e., $I(\mathcal{A}(\mathcal{D}_\theta); \mathcal{D}_\theta)$. For practical learning algorithms with bounded capacity, if $\mathcal{A}$ always outputs a model $h$ with at most $N$ parameters each represented by a $b$-bit floating number, then $I(\mathcal{A}(\mathcal{D}_\theta); \mathcal{D}_\theta) \leq bN$ by information theory. Empirically, Allen-Zhu and Li [2024a] found that $I(\mathcal{A}(\mathcal{D}_\theta); \mathcal{D}_\theta) \approx 2N$ holds across various training setups by controlled experiments.

We model a sufficiently trained LM with capacity $M$ as an optimal bounded-capacity learner, which minimizes the expected loss as much as possible under the capacity constraint $M$:

**Definition 4.1** (Optimal Bounded-Capacity Learner). For a given prior $\mathcal{P}$ and $M > 0$, the best achievable loss under the capacity constraint $M$ is defined as

$$F_\mathcal{P}(M) := \inf_\mathcal{A} \left\{ \bar{\mathcal{L}}_\mathcal{P}(\mathcal{A}) : I(\mathcal{A}(\mathcal{D}_\theta); \mathcal{D}_\theta) \leq M \right\}, \tag{2}$$

where the infimum is taken over all learning algorithms. An optimal $M$-bounded-capacity learner is a learning algorithm $\mathcal{A}$ such that $I(\mathcal{A}(\mathcal{D}_\theta); \mathcal{D}_\theta) \leq M$ and $\bar{\mathcal{L}}_\mathcal{P}(\mathcal{A}) = F_\mathcal{P}(M)$.

## 4.3 Warmup: Training Exclusively on Mixture of Facts

We start with a simple case where the data distribution $\mathcal{D}_\theta$ contains $K$ random facts. Each fact is a pair $(X_i, y_i)$, where $X_i$ is a set of input contexts (e.g., paraphrases) and $y_i$ is the target token. For

instance, the fact "Gracie Tessa Howell was born in 1946" can have contexts like "Gracie Tessa Howell's birth year is" or "Gracie Tessa Howell came to this world in the year," all mapping to the target $y$ = "1946". We further assume that $X_1, \ldots, X_K$ are disjoint.

Let $\mathcal{D}_\theta(y \mid x)$ be the next-token distribution given context $x$. The universe samples $y_1, y_2, \ldots, y_K$ independently from fixed distributions $\mathcal{Y}_1, \ldots, \mathcal{Y}_K$ and sets $\theta = (y_1, \ldots, y_K)$. The universe further sets $\mathcal{D}_\theta(y \mid x_i)$ as a point mass at $y_i$, $\forall x_i \in X_i$. Other inputs $x$ may occur in $\mathcal{D}_\theta$, but their target tokens are independent of $\theta$. Define the exposure frequency of the $i$-th fact as the total probability that any $x \in X_i$ appears in $\mathcal{D}_\theta$: $\sum_{x' \in X_i} \mathbb{P}_\theta(x = x')$. If all $K$ facts have equal exposure frequency $p$ (despite different entropies), a bounded-capacity learner reduces expected loss *linearly* with capacity $M$, thus no phase transitions:

**Theorem 4.2.** *For all $M \geq 0$, if all the facts have the same exposure frequency $p$, then*

$$F_\mathcal{P}(M) = C + p \cdot \max\{H_{\text{tot}} - M, 0\}, \tag{3}$$

*where $H_{\text{tot}} := \sum_{i=1}^K H(\mathcal{Y}_i)$ and $C := F_\mathcal{P}(\infty)$.*

### 4.4 Data Mixing Induces Phase Transitions

*What if we mix the random facts with data from another domain, say web text?* Consider a data distribution $\mathcal{D}_\theta$ composed of two domains: (1) a mixture of $K$ random facts (as in Section 4.3) and (2) another domain with a much more complex structure. Let the latent variable $\theta = (\theta_1, \theta_2)$, where $\theta_1$ governs the distribution of $K$ random facts, $\mathcal{D}_{\theta_1}^{(1)}$, and $\theta_2$ governs the data distribution of the second domain, $\mathcal{D}_{\theta_2}^{(2)}$. Assume the universe draws $\theta_1$ and $\theta_2$ independently from priors $\mathcal{P}_1$ and $\mathcal{P}_2$, respectively. The overall data distribution $\mathcal{D}_\theta$ is $\mathcal{D}_\theta = r\mathcal{D}_{\theta_1}^{(1)} + (1-r)\mathcal{D}_{\theta_2}^{(2)}$, with mixing ratio $r \in (0, 1)$. Let $p$ denote the exposure frequency of each fact in $\mathcal{D}_{\theta_1}^{(1)}$, and $H_{\text{tot}} := \sum_{i=1}^K H(\mathcal{Y}_i)$ be the total entropy of the target tokens in the first domain (as in Section 4.3). For simplicity, we assume the two domains contain non-overlapping information (see Definition F.5).

To measure models' performance on the first domain after training with algorithm $\mathcal{A}$ on the data mixture, we define $\bar{\mathcal{L}}_1(\mathcal{A}) := \mathbb{E}_{\theta \sim \mathcal{P}_1}[\mathcal{L}(\mathcal{A}(\mathcal{D}_\theta); \mathcal{D}_{\theta_1}^{(1)})]$ as the model's expected loss on the first domain. If $\bar{\mathcal{L}}_1(\mathcal{A}) = F_{\mathcal{P}_1}(0)$, then the model learns nothing (random guessing). If $\bar{\mathcal{L}}_1(\mathcal{A}) = F_{\mathcal{P}_1}(\infty)$, the model perfectly learns the facts.

Theorem 4.3 shows that the learner sharply transitions between the two extremes as model size increases. This transition is characterized by two functions: $M_0^-(t) := \sup\{M \geq 0 : -F'_{\mathcal{P}_2}(M) > t\}$ and $M_0^+(t) := \inf\{M \geq 0 : -F'_{\mathcal{P}_2}(M) < t\}$. By rate-distortion theorem, $F_{\mathcal{P}_2}(M)$ is convex and hence $-F'_{\mathcal{P}_2}(M)$ is non-increasing. Thus, $M_0^-(t)$ and $M_0^+(t)$ mark the last and first model sizes where $-F'\mathcal{P}_2(M)$ exceeds or falls below $t$. If $F'_{\mathcal{P}_2}(M)$ is strictly decreasing, then $M_0^-(t) = M_0^+(t)$.

**Theorem 4.3** (Phase Transition in Model Size)**.** *For any optimal $M$-bounded-capacity learner $\mathcal{A}$,*

1. *if $M \leq M_0^-(\frac{r}{1-r} \cdot p)$, then $\bar{\mathcal{L}}_1(\mathcal{A}) = F_{\mathcal{P}_1}(0)$;*

2. *if $M \geq M_0^+(\frac{r}{1-r} \cdot p) + H_{\text{tot}}$, then $\bar{\mathcal{L}}_1(\mathcal{A}) = F_{\mathcal{P}_1}(\infty)$.*

**Key Example: When Web Data Loss Follows a Power Law in Model Size.** Consider the case where $F_{\mathcal{P}_2}(M)$ is a power-law function of $M$, *i.e.*, $F_{\mathcal{P}_2}(M) = C + A \cdot M^{-\alpha}$. Here, $\alpha \in (0, 1)$ and $A$ is a large constant. This is a reasonable assumption since LLM pre-training usually exhibits such power-law scaling behavior in model size [Kaplan et al., 2020, Hoffmann et al., 2022]. In this case, taking the derivative of $F_{\mathcal{P}_2}(M)$ gives $-F'_{\mathcal{P}_2}(M) = A \cdot \alpha \cdot M^{-\alpha-1}$. Then, $M_0^-(t) = M_0^+(t) = (\frac{A\alpha}{t})^{1/(\alpha+1)}$. Plugging this into Theorem 4.3, we have the critical value for model size:

$$M_{\text{thres}} \sim \left(\frac{1}{rp}\right)^{1/(\alpha+1)}. \tag{4}$$

This implies that a small $r$ or $p$ may cause the model to learn nothing from the knowledge-dense dataset, even if its capacity is sufficient to learn the entire dataset.

Arranging the terms in (4), we can also obtain the critical value in the mixing ratio $r$:

$$r_{\text{thres}} \sim \frac{1}{p \cdot M^{\alpha+1}}. \tag{5}$$

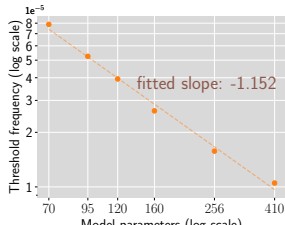

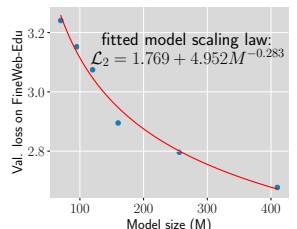

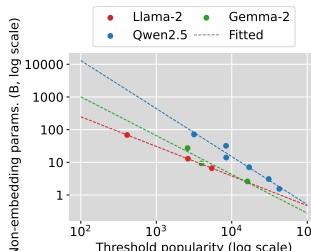

(a) Threshold frequency of synthetic biographies across different model sizes.

(b) The scaling law for the validation loss on FineWeb-Edu with respect to model size.

(c) The threshold popularity for knowledge tested in PopQA v.s. model size.

Figure 7: Validating the power-law relationship of threshold Frequency and model size. (a) & (b): Experiments on the mixture of SynBio-10k-power-law and FineWeb-Edu confirm that (1) the threshold frequency follows a power-law relationship with model size, and (2) the power-law exponent is approximately equal to the model scaling exponent plus one. (c): For the three open-source model families we examined, the threshold popularity for knowledge tested in PopQA also follows a power-law relationship with model size.

**Threshold Frequency for a Single Fact.** For each fact in the first domain, its overall probability of being sampled is $rp$ in the data mixture. Again, arranging the terms in (5), we obtain that for a single fact to be learned by the model, its frequency of appearing in the pre-training corpus should be larger than a threshold frequency $f_{\text{thres}}$, which scales with the model size following a power law:

$$f_{\text{thres}} \sim \frac{1}{M^{\alpha+1}}. \tag{6}$$

## 5 Power-Law Relationship of Threshold Frequency and Model Size

In this section, we validate the predicted power-law relationship between model size and threshold frequency on both synthetic biographies and a set of knowledge extracted from Wikipedia.

### 5.1 Experiments on Synthetic Biographies

We construct SynBio-10k-power-law, where 10k biographies are divided into 100 subsets of 100 individuals, with subset sampling probability following a power-law distribution (exponent 1.5). Within each subset, all biographies have a uniform sampling probability. We then mix this dataset with FineWeb-Edu using $r = 0.01$ and train models under this setup.

To estimate the threshold frequency $f_{\text{thres}}$, we sort the subsets by sampling probability in descending order and identify the first group where model accuracy falls below a target value $\alpha_{\text{target}}$. The frequency of biographies in this subset is used to approximate $f_{\text{thres}}$. We use $\alpha_{\text{target}} = 80\%$.

As shown in Figure 7(a), $\log f_{\text{thres}}$ and $\log M$ exhibit a linear relationship, yielding a slope of $1.152$. This value is larger than 1, as expected from our theory. Further, we wonder if this slope is indeed close to $\alpha + 1$. Following the approach of Hoffmann et al. [2022], we fit a model scaling function for FineWeb-Edu validation loss in Figure 7(b), obtaining $\alpha \approx 0.283$. This leads to a predicted exponent of $1.283$, which is close to the observed value of $1.152$.

### 5.2 Experiments on Knowledge Extracted from Wikipedia

We further evaluate models on PopQA [Mallen et al., 2023], which contains 14k QA pairs derived from Wikidata triplets, along with monthly page view for corresponding Wikipedia articles. Since knowledge tested in PopQA can be structured as triplets, we consider them as homogeneous and expect them to exhibit similar threshold frequencies for a given model size.

**Estimating the Threshold Frequency.** Counting the frequency of specific knowledge in the pre-training data is challenging due to the scale [Kandpal et al., 2023]. Following Mallen et al. [2023], we use Wikipedia page views as a proxy for popularity, which is assumed roughly proportional to the frequency of the knowledge in web data. To estimate the threshold popularity $P_{\text{thres}}$, we identify the smallest popularity $P$ such that the model's accuracy on knowledge with popularity above $P$ meets the target accuracy $\alpha_{\text{target}}$ which is set to 60% in our experiments. See Appendix E.6 for details.

**Threshold frequency and model size follow a power law.** We examine base models from Llama-2 [Touvron et al., 2023], Qwen-2.5 [Qwen et al., 2024], and Gemma-2 [Team et al., 2024], which

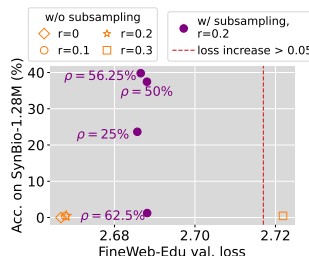
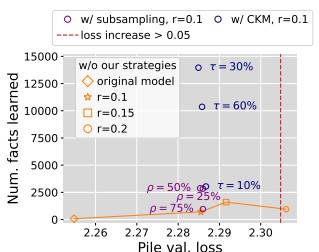
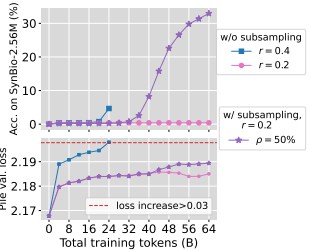

(a) 410M, train from scratch on FineWeb-Edu&SynBio-1.28M

(b) 410M, continual pre-train on the Pile&WikiBio.

(c) 1B, continual pre-train on the Pile&SynBio-2.56M.

Figure 8: Our proposed strategies significantly boost knowledge acquisition under low mixing ratios while preserving models' general capability.

are likely trained on similar data mixtures within each family. Figure 7(c) reveals that $\log P_{\text{thres}}$ generally decreases linearly as $\log$ model size increases, though the slope varies across families due to differences in architecture and training data. We examine more model families in Appendix D.4.

# 6 Strategies to Enhance Knowledge Acquisition Under Low Mixing Ratios

Inspired by our theory, we propose two simple yet effective strategies to enhance knowledge acquisition under low mixing ratios. This setting is common in practice, as a large $r$ may harm general capabilities expected to be acquired from other data sources. The key idea is to raise the frequency of each fact, thereby increasing the "marginal value" of the knowledge-dense data.

- **Strategy 1: Random Subsampling**: Randomly subsample the knowledge dataset.
- **Strategy 2: Compact Knowledge Mixing (CKM)**: Rephrase the knowledge into a compact form and add the rephrased version to the original dataset while keeping the overall mixing ratio fixed. See implementation details in Appendix E.7.

We validate these strategies on SynBio and WikiBio, a curated dataset of Wikipedia biographies. For example, on WikiBio, random subsampling and CKM improve the number of learned facts by 4x and 20x, respectively. The effectiveness of random subsampling is especially surprising, as it yields higher accuracy despite discarding a significant proportion of the knowledge-dense data.

## 6.1 Real-World Knowledge Data: WikiBio

To extend our study to a more real-world scenario, we curate WikiBio, a dataset containing Wikipedia biographies along with ten paraphrased versions for 275k individuals, totaling 453M tokens. This task is more challenging than SynBio as WikiBio features diverse texts without uniform formats, requiring the model to generalize to queries that rarely have exact matches in the training data. See Appendix E.2.2 for dataset construction details and Appendix E.7 for evaluation details.

## 6.2 Strategy 1: Random Subsampling

While random subsampling seems counterintuitive at first glance, it becomes reasonable if we consider how the threshold mixing ratio $r_{\text{thres}}$ relates to the exposure frequency of each fact within the knowledge-dense dataset, denoted as $p$. For a dataset containing only $S$ facts with uniform probability, $p \propto 1/S$. We can derive from (5) that the threshold mixing ratio $r_{\text{thres}} \sim \frac{S}{M^{\alpha+1}}$. Subsampling reduces $S$ and thus lowers the threshold mixing ratio, allowing the model to achieve much higher accuracy on the subsampled dataset. We use $\rho$ to represent the subsampling ratio below.

**Experimental Setup.** We study both pre-training from scratch and continual pre-training setups. To evaluate the model's general capabilities, we use its validation loss on the web data (the Pile or FineWeb-Edu) and its zero-shot performance on five downstream tasks (see details in Appendix D.7). We compare the validation loss and average downstream performance to the model trained with $r = 0$ in the pre-training-from-scratch setup or to the original Pythia model in the continual pre-training setup. Downstream performance drop of more than 2% is considered unacceptable.

**Subsampling enables faster knowledge acquisition while maintaining general capability.** We train 410M models from scratch FineWeb-Edu mixed with SynBio-1.28M using $r \in \{0, 0.1, 0.2, 0.3\}$

for a total of 32B tokens. As shown in Figures 8(a) and 14(a), increasing $r$ degrades FineWeb-Edu validation loss and downstream accuracy, with performance becoming unacceptable at $r = 0.3$ (-2.09% accuracy, +0.05 loss), while SynBio accuracy remains near zero. In contrast, subsampling SynBio-1.28M to 25%, 50%, and 56.25% boosts SynBio accuracy to 23.53%, 37.46%, and 39.81%, respectively, while maintaining downstream performance within the acceptable range. Note that further increasing $\rho$ to 62.5% makes the frequency of each biography too low, resulting in SynBio accuracy dropping back to near zero. See more details in Appendix E.7, Tables 2(b) and 3(a).

**Consistent Results for Continual Pre-training.** We continually pre-train the 410M or 1B Pythia models from their 100k-step checkpoints on the mixture of the Pile and WikiBio or SynBio-2.56M. The 410M models are trained for 32B tokens and 1B models for 64B. When $r$ is large, the Pile validation loss may increase with training due to catastrophic forgetting [Ibrahim et al., 2024]. To preserve models' general capabilities, we apply early stopping when Pile validation loss increases by 0.05 (410M model) or 0.03 (1B model), each corresponding to $\sim 2\%$ drop in downstream performance. As shown in Figures 8(b) and 14(b), without subsampling, $r = 0.1$ or 0.15 results in slow learning of WikiBio, while $r = 0.2$ triggers early stopping after 20B tokens, resulting in poor WikiBio performance. By contrast, subsampling WikiBio to 25% or 50% significantly accelerates knowledge acquisition and keeps Pile validation loss acceptable. For example, for $r = 0.1$, setting $\rho$ to 50% improves the number of learned facts by 4 times. Similar trends hold for 1B models: subsampling SynBio to 50% at $r = 0.2$ outperforms both $r = 0.2$ and early-stopped $r = 0.4$ without subsampling by $\sim 30\%$. See more details in Appendix E.7, Tables 2(c), 3(b) and 4.

### 6.3 Strategy 2: Compact Knowledge Mixing (CKM)

The second strategy rephrases knowledge into compact forms (e.g., tuples) and adds them to the original dataset. Given that the frequency of each fact $f$ is inversely proportional to its average representation token count, this augmentation reduces the average token count, thereby increasing $f$'s effective frequency and potentially pushing it above the threshold $f_{\text{thres}}$. CKM is in the same spirit as the data synthesis technique in Su et al. [2024], which rephrases high-quality data into condense forms such as QA pairs and knowledge lists.

For WikiBio, we compress the key information—name, birth date, and occupation—into the tuple format "Bio: N {name} B {birth date} O {occupation}". We add these tuples until their token count equals a proportion $\tau$ (which we call the CKM ratio) of the original dataset's token count.

**Experimental Setup.** We apply CKM to WikiBio with the same continual pre-training setup as in Section 6.2. We apply early stopping when Pile validation loss increases by 0.05.

**CKM significantly improves knowledge acquisition efficiency while preserving general capability.** We fix $r = 0.1$ and explore CKM ratios $\tau \in \{0.1, 0.3, 0.6\}$, which correspond to roughly 2x, 3x, and 4x increases in fact frequency, respectively. As shown in Figures 8(b) and 14(c), CKM preserves the general capability and consistently boosts knowledge acquisition. Notably, performance on WikiBio improves by 4x when the short-form augmentation makes up only 10% tokens of the original WikiBio dataset. Increasing $\tau$ to 30% further boosts the number of learned facts by 20x. See downstream performance in Table 5.

## 7 Discussions and Future Directions

**Extensions to reasoning tasks.** Although our experiments mainly investigate factual knowledge, we also identify phase transitions in simple reasoning tasks. This suggests a commonality: memorization is foundational to reasoning, not just to fact-recall. Without basic knowledge, models cannot reason effectively, an observation shared by Ruis et al. [2024], Xie et al. [2024]. For instance, solving math problems requires memorizing theorems, definitions, and techniques. We defer the exploration of more complex reasoning tasks to future work.

**Connection to real-world data.** Following Allen-Zhu and Li [2024a], we use synthetic biographies as a proxy for knowledge-dense data for controlled and quantitative experiments. In contrast, real-world datasets are more heterogeneous—for example, Wikipedia includes both simple facts (e.g., biographies) and more complex content (e.g., scientific theories). These types of knowledge vary in learning difficulty and may exhibit different threshold frequencies. As a result, phase transitions may not be as apparent when mixing a heterogeneous dataset with web text. Nevertheless, our findings still apply at the level of individual knowledge pieces. That is, a specific fact or reasoning procedure may not be acquired at all if its frequency or the model size falls below a threshold, as evidenced by the theoretical result in (6) and empirical evidence in Section 5 and Section 3.3.

## Acknowledgements and Disclosure of Funding

J.Z. acknowledges support by the National Key R&D Program of China 2024YFA1015800 and Shanghai Qi Zhi Institute Innovation Program.

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

# Contents

# A  Limitations

The high computational costs to conduct all these experiments impede us from replicate all the experiments with different random seeds. These costs include the number of GPU hours. For example, a typical run of training a 410M model for 32B tokens requires 256 A100 GPU hours. Despite these difficulties, we managed to conduct experiments on models up to 6.9B and conduct ablation studies on hyperparameters in Appendix D.2.

# B  Broader Impact

This paper identifies two phase transitions in knowledge acquisition within data mixtures and provides theoretical understanding of these phenomena. Building on our theory, we propose two strategies to enhance the efficiency of knowledge acquisition. Our findings offer deeper insights into LLM behavior and can be applied to improve the factual accuracy of LLMs.

# C  Related Works

**Knowledge Capacity Scaling Law.**  LLMs are typically trained on a vast amount of data that are rich in knowledge, and extensive studies have investigated how much knowledge LLMs can acquire from the training data. Pioneering studies [Petroni et al., 2019, Roberts et al., 2020, Da et al., 2021] demonstrate that LLMs can capture a substantial amount of knowledge, suggesting their potential as knowledge bases. To quantify the relationship between model size and knowledge storage, Allen-Zhu and Li [2024a] and Lu et al. [2024] discover a linear relationship between models' knowledge capacity and their parameter count by training LLMs on data only containing fixed-format knowledge for sufficiently long horizons. Later, Nichani et al. [2025] formally proved this linear relationship. In contrast, this paper examines the data mixing scenario and demonstrates that this linear scaling can be disrupted when the knowledge-dense dataset is mixed with vast amounts of web-scraped data. Another important factor is the frequency of occurrence for knowledge.

**Impact of Frequency on Knowledge Acquisition.**  This paper identifies phase transitions in knowledge acquisition within data mixtures with respect to model size and mixing ratio. Some relevant observations can be found in previous papers, but we takes a more direct and systematic approach. Kandpal et al. [2023], Mallen et al. [2023], Sun et al. [2024] find that LLMs can perform poorly on low-frequency knowledge. Ghosal et al. [2024] show that frequency of knowledge in the pre-training data determines how well the model encodes the knowledge, which influences its extractability after QA fine-tuning. Taking a more microscopic view, Chang et al. [2024] insert a few pieces of new knowledge during training and track their loss. By fitting a forgetting curve, they conjecture that the model may fail to learn the knowledge if its frequency is lower than some threshold.

**Memorization and Forgetting.**  Our findings also relate to prior observations on the memorization and forgetting behaviors of LLMs, but we explicitly characterize phase transitions in the context of data mixing. Carlini et al. [2023] show that memorization of training data follows a log-linear relationship with model size, the number of repetitions, and prompt length. Biderman et al. [2024] take a data point-level perspective and demonstrate that it is difficult to predict whether a given data point will be memorized using a smaller or partially trained model. By injecting a few new sequences into the training data, Huang et al. [2024] find that a sequence must be repeated a non-trivial number of times to be memorized. By examining training dynamics, Tirumala et al. [2022] observe that memorization can occur before overfitting and that larger models memorize faster while forgetting more slowly. Zucchet et al. [2025] study the training dynamics governing factual knowledge acquisition of LLMs and find that the performance can undergo a plateau before the model acquires precise knowledge, during which the attention-based circuits form. From a theoretical perspective, Feldman [2020] prove that memorization of training labels is necessary to achieve near-optimal generalization error for long-tailed data distributions.

**Scaling laws for Data Mixing.**  LLM performance is significantly influenced by the mixing proportions of the training data from different domains. Our paper is related to a line of studies that optimize the mixing proportions by modeling LLM performance as a function of the mixing proportions [Liu et al., 2024, Kang et al., 2024, Ye et al., 2024, Ge et al., 2024]. However, their datasets can be highly

heterogeneous even within a single domain (e.g., OpenWebText, Pile-CC) while we focus on mixing a uniform, knowledge-dense dataset into web-scraped data.

# D  Additional Experimental Results

## D.1  Additional Results for Phase Transitions

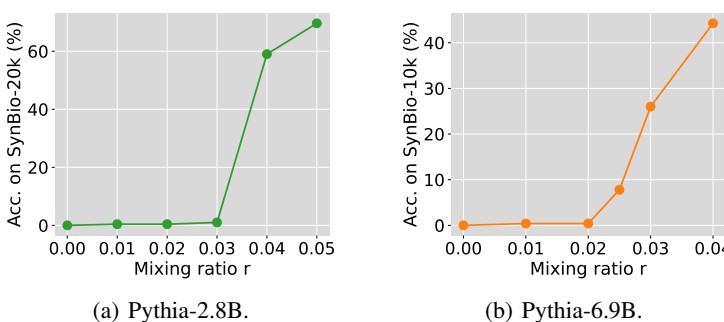

(a) Pythia-2.8B.  (b) Pythia-6.9B.

Figure 9: Phase transition in mixing ratio persists for larger models. We train Pythia-2.8B and Pythia-6.9B with 2B and 1B total training tokens, respectively. To ensure sufficient exposure to SynBio within these training horizons, we use smaller SynBio datasets—SynBio-20k for the 2.8B model and SynBio-10k for the 6.9B model—mixed with FineWeb-Edu.

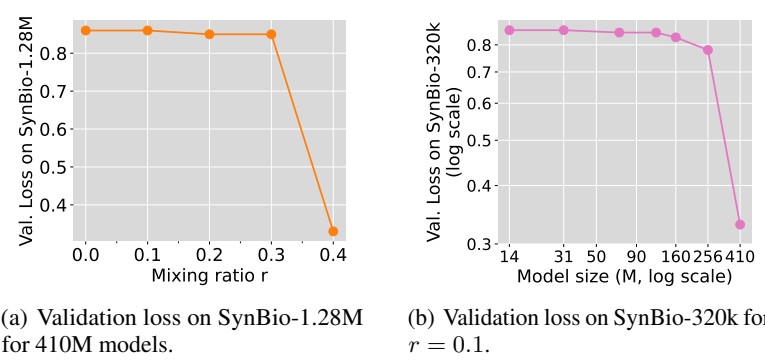

(a) Validation loss on SynBio-1.28M for 410M models.  (b) Validation loss on SynBio-320k for $r = 0.1$.

Figure 10: In addition to discrete metrics like accuracy, we can also observe phase transitions in validation loss, a continuous metric.

Table 1: We replicate the experiments in Figure 2(a) with three different random seeds and report the mean and standard deviation below. While accuracy varies slightly with random seeds, the phase transition behavior remains consistent and clearly observable across runs.

| $r$ | Mean Acc. (%) | Std. Dev. (%) |
|------|------|------|
| 0.1 | 0.4 | 0.0 |
| 0.15 | 0.4 | 0.0 |
| 0.2 | 0.4 | 0.0 |
| 0.25 | 0.8 | 0.0 |
| 0.3 | 7.3 | 2.8 |
| 0.35 | 27.5 | 2.8 |
| 0.4 | 40.4 | 3.1 |
| 0.45 | 58.1 | 3.1 |

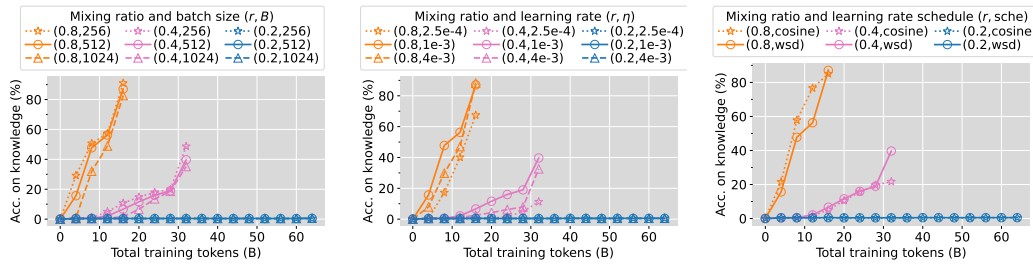

(a) Vary the batch size.

(b) Vary the peak learning rate.

(c) Vary the learning rate schedule. Both schedules use a peak learning rate of $10^{-3}$.

Figure 11: Ablation studies on hyperparameters. The models exhibit consistent trends in knowledge acquisition across different batch sizes, learning rate values and schedules. All experiments are conducted by training 70M models on the mixture of FineWeb-Edu and SynBio-320k.

## D.2 Ablation Studies

We now conduct ablation studies to demonstrate the robustness of our findings with respect to hyperparameters. We explore $r \in \{0.2, 0.4, 0.8\}$ and train 70M models for a total of 64B, 32B, and 16B tokens, respectively, ensuring each configuration passes SynBio the same number of times.

**Consistent Trends Across Different Batch Sizes.** As shown in Figure 11(a), we evaluate three batch sizes, $B \in \{256, 512, 1024\}$, for each $r$ and observe consistent general trends across all batch sizes. For $r = 0.4$ and $r = 0.8$, smaller batch sizes yield slightly higher accuracies, likely due to the increased number of update steps. These experiments further distinguish between two types of frequency at which the model encounters the knowledge dataset: per-token frequency and per-step frequency. For a fixed mixing ratio, doubling the batch size doubles the occurrences of each biography per step, while the occurrences per token remain unchanged. The results demonstrate that per-token frequency, rather than per-step frequency, determines training efficiency in knowledge acquisition.

**Consistent trends across learning rate values and schedules.** In Figure 11(b), we explore peak learning rates among $\{2.5 \times 10^{-4}, 10^{-3}, 4 \times 10^{-3}\}$ using the WSD scheduler. We observe that the trends are consistent across these values, although the learning process slows down at the lowest value $2.5 \times 10^{-4}$. In Figure 11(c), results for both cosine and WSD schedulers show similar trends.

## D.3 Additional Discussions and Experiments on Reasoning Tasks

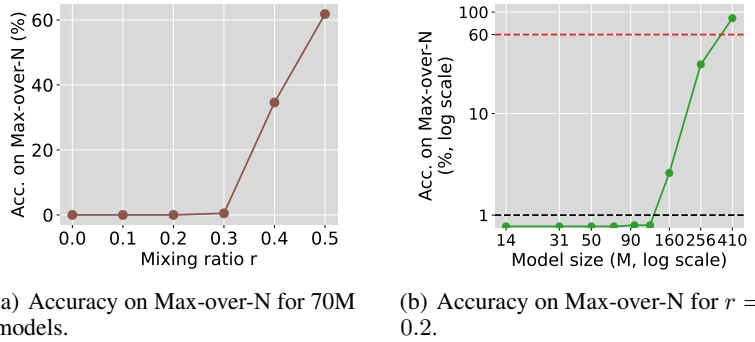

(a) Accuracy on Max-over-N for 70M models.

(b) Accuracy on Max-over-N for $r = 0.2$.

Figure 12: Phase transitions in the Max-over-N task. Here, we set $N = 30$.

**Discussion on the slope calculation task: model is indeed learning the procedure rather than memorizing the training data.** We show that the model cannot rely purely on memorization to

solve the slope calculation tasks specified in Section 3.3. Specifically, the total number of possible slope calculation problems in our setup is $100 \times 100 \times 99 \times 100 = 9.9 \times 10^7$. In contrast, a typical training run in Figure 5 with $r = 0.4$ sees fewer than $3.5 \times 10^6$ unique slope calculation examples, less than 5% of the full space. Despite this limited coverage, the model still achieves 60% accuracy at test time, where examples are uniformly sampled from the full distribution. This substantial generalization beyond the training data suggests that the model is indeed learning a generalizable computation procedure, rather than memorizing specific input-output pairs.

**Experiments on Another Reasoning Task with Larger Input Space: Max-over-N.** The slope calculation task suggests that the model learns a generalizable procedure. To test this hypothesis in a more challenging setting, we introduce another task named "Max-over-N". This task is explicitly designed with a vastly larger input space, rendering memorization computationally infeasible. Specifically, the model is asked to find the maximum number given a list of $N = 30$ integers, each randomly sampled from $\{0, 1, \cdots, 99\}$. This creates an enormous input space of $10^{60}$. The format of the training samples is shown in Table 13.

Following the setup in Section 3.3, we add 3M tokens of such examples to the OpenWebMath dataset (accounting for less than 0.02% of the original OpenWebMath token count) to create a modified version. We then train Pythia models on the mixture of FineWeb-Edu and the modified OpenWebMath dataset, with $r$ denoting the mixing ratio of the modified OpenWebMath. For evaluation, we generate 1,000 test samples of "Max-over-N" and assess whether the model outputs the correct final answer.

As shown in Figure 12, we observe the same phase transition phenomena with respect to both model size and mixing ratio, consistent with our main findings. The key result is that the 70M model achieves 60% test accuracy after training on fewer than 3,500 unique examples (at $r = 0.5$). This training set is an infinitesimal fraction of the $10^{60}$ possible inputs. This clearly demonstrates generalization beyond memorization.

## D.4 Additional Results for Validating the Power-Law Relationship of Threshold Frequency and Model Size

In Figure 13, we relax the constraint of training on the same data mixture and investigate the overall trend between model size and $P_{\text{thres}}$. We add the Llama-3 [Dubey et al., 2024] family, and evaluate both base and instruction-tuned models for all families, totaling 30 models. Interestingly, in Figure 13, $\log$ model size and $\log P_{\text{thres}}$ also exhibit a linear relationship, with most models falling within the 95% confidence interval. We further use models from the OLMo [Groeneveld et al., 2024] family as a validation set, where predictions of the fitted power law closely match the ground truth.

**Potential Application: Inferring the Size of Proprietary Models.** The identified power-law relationship offers a potential method for estimating the size of proprietary models, such GPTs. As a preliminary attempt, we estimate the threshold popularity for GPT-3.5-Turbo, GPT-4, GPT-4o, and GPT-4o-mini. Applying the fitted power law yields size predictions of 61B, 514B, 226B, and 24B, respectively. The 95% confidence intervals are 12–314B, 80–3315B, 39–1313B, and 5–118B, respectively.

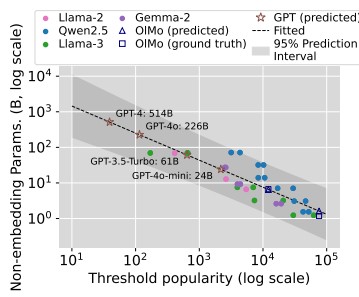

Figure 13: For 410M models trained on the mixture of FineWeb-Edu and SynBio-1.28M, accuracy for $r = 0.2$ remains near zero even when we extend the training by 4 times.

## D.5 Additional Plots for Strategies to Enhance Knowledge Acquisition

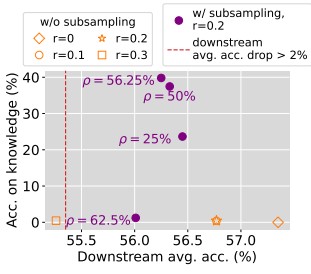

(a) 410M, trained from scratch on the mixture of FineWeb-Edu and SynBio-1.28M.

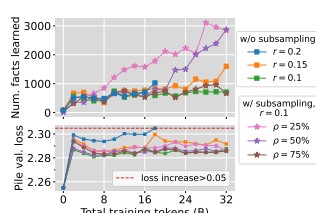

(b) Training trajectory for applying subsampling to WikiBio.

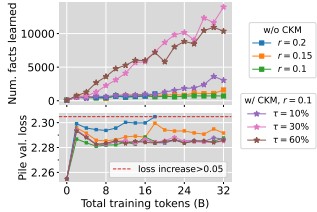

(c) Training trajectory for applying CKM to WikiBio.

Figure 14: Additional plots for strategies to enhance knowledge acquisition.

## D.6 Detailed Performance on SynBio

In Table 2(a), we detail the accuracy of each attribute for 70M models trained on the mixture of FineWeb-Edu and SynBio-320k with $r \in \{0.2, 0.4, 0.8\}$, trained for 64B, 32B, and 16B tokens respectively. We notice that the accuracy for birth date is lower than other attributes. This can be attributed to the complexity of precisely recalling the combined elements of day, month, and year information, which together form a much larger domain than other attributes. To maintain clarity and conciseness, we omit the detailed performance in other 70M experiments, as this pattern persists across them.

Furthermore, we present the detailed performance of 410M models on SynBio-1.28M corresponding to Figure 8(a) in Table 2(b). We also provide the detailed performance of 1B models on SynBio-2.56M corresponding to Figure 8(c) in Table 2(c).

Table 2: Detailed performance on SynBio. We report the accuracy (%) for each attribute averaged over five templates.

(a) 70M model, pre-trained from scratch on the mixture of FineWeb-Edu and SynBio-320k.

| $r$ | Birth date | Birth city | University | Major | Employer | Avg. |
|---|---|---|---|---|---|---|
| Random guess | 0.00 | 0.50 | 0.33 | 1.00 | 0.38 | 0.44 |
| 0.2 | 0.00 | 0.63 | 0.43 | 1.12 | 0.38 | 0.51 |
| 0.4 | 16.96 | 45.67 | 41.03 | 50.78 | 43.93 | 39.68 |
| 0.8 | 79.76 | 88.64 | 88.55 | 90.10 | 88.30 | 87.07 |

(b) 410M model, pre-trained from scratch on the mixture of FineWeb-Edu and SynBio-1.28M.

| $N$ | $\rho$ (%) | $r$ | Birth date | Birth city | University | Major | Employer | Avg. |
|---|---|---|---|---|---|---|---|---|
| Random guess | | | 0.00 | 0.50 | 0.33 | 1.00 | 0.38 | 0.44 |
| - | - | 0.00 | 0.00 | 0.00 | 0.00 | 0.00 | 0.00 | 0.00 |
| 1.28M | 100 | 0.1 | 0.00 | 0.42 | 0.33 | 1.01 | 0.21 | 0.39 |
| 1.28M | 100 | 0.2 | 0.00 | 0.45 | 0.34 | 1.09 | 0.22 | 0.42 |
| 1.28M | 100 | 0.3 | 0.00 | 0.49 | 0.35 | 1.14 | 0.25 | 0.45 |
| 320k | 25 | 0.2 | 22.34 | 23.98 | 23.64 | 24.03 | 23.65 | 23.53 |
| 640k | 50 | 0.2 | 27.97 | 39.66 | 38.51 | 41.50 | 39.68 | 37.46 |
| 720k | 56.25 | 0.2 | 28.02 | 42.94 | 42.15 | 44.07 | 41.88 | 39.81 |
| 800k | 62.5 | 0.2 | 0.01 | 1.16 | 0.85 | 3.19 | 0.89 | 1.22 |

(c) 1B model, continually pre-trained on the mixture of the Pile and SynBio-2.56M. Note that $r = 0.4$ is early stopped due to its Pile validation loss increasing beyond the acceptable range.

| $N$ | $\rho$ (%) | $r$ | Training tokens (B) | Birth date | Birth city | University | Major | Employer | Avg. |
|---|---|---|---|---|---|---|---|---|---|
| Random guess | | | | 0.00 | 0.50 | 0.33 | 100 | 0.38 | 0.44 |
| Pythia-1B-100k-ckpt | | | | 0.00 | 0.00 | 0.00 | 0.00 | 0.00 | 0.00 |
| 2.56M | 100 | 0.2 | 64 | 0.01 | 0.46 | 0.33 | 0.98 | 0.21 | 0.39 |
| 2.56M | 100 | 0.4 | 24 | 0.05 | 10.95 | 3.90 | 4.74 | 3.64 | 4.66 |
| 1.28M | 50 | 0.2 | 64 | 23.95 | 34.55 | 35.05 | 35.96 | 35.19 | 32.94 |

## D.7 Detailed Downstream Performance

We employ the `lm-eval-harness` [Gao et al., 2024] codebase to evaluate the zero-shot performance on five downstream tasks, including LAMBADA [Paperno et al., 2016], ARC-E [Clark et al., 2018], PIQA [Bisk et al., 2020], SciQ [Welbl et al., 2017], and HellaSwag [Zellers et al., 2019], covering core capabilities such as text understanding, commonsense reasoning, and question answering. We compute the validation loss on about 50M tokens on a holdout set from the Pile or FineWeb-Edu. The detailed downstream performance and validation loss for applying the random subsampling

strategy to SynBio and WikiBio are presented in Tables 3 and 4, respectively. Additionally, we report the detailed downstream results for applying CKM to WikiBio in Table 5.

Table 3: Detailed downstream performance and validation loss for applying the random subsampling strategy to SynBio. We report the accuracy (%) and standard deviation (%) in the format $\text{acc.}_{\text{(std. dev.)}}$ for each downstream task.

(a) 410M model, train from scratch.

| $N$ | $\rho\,(\%)$ | $r$ | LAMBADA | ARC-E | Sciq | PIQA | HellaSwag | Avg. | FineWeb-Edu val. loss |
|---|---|---|---|---|---|---|---|---|---|
| - | - | 0 | $38.25_{(0.68)}$ | $61.83_{(1.00)}$ | $83.60_{(1.17)}$ | $68.01_{(1.09)}$ | $35.04_{(0.48)}$ | 57.35 | 2.667 |
| 1.28M | 100 | 0.1 | $34.56_{(0.66)}$ | $62.33_{(0.99)}$ | $83.50_{(1.17)}$ | $68.34_{(1.09)}$ | $35.13_{(0.48)}$ | $56.77(\downarrow 0.58)$ | $2.668(\uparrow 0.001)$ |
| 1.28M | 100 | 0.2 | $34.43_{(0.67)}$ | $62.13_{(0.99)}$ | $83.80_{(1.17)}$ | $68.12_{(1.09)}$ | $35.39_{(0.48)}$ | $56.77(\downarrow 0.58)$ | $2.668(\uparrow 0.001)$ |
| 1.28M | 100 | 0.3 | $33.94_{(0.66)}$ | $60.77_{(1.00)}$ | $80.80_{(1.25)}$ | $66.54_{(1.10)}$ | $34.23_{(0.47)}$ | $55.26(\downarrow 2.09)$ | $2.722(\uparrow 0.054)$ |
| 320k | 25 | 0.2 | $36.70_{(0.67)}$ | $60.35_{(1.00)}$ | $(82.70_{1.20})$ | $67.74_{(1.09)}$ | $34.76_{(0.48)}$ | $56.45(\downarrow 0.90)$ | $2.686(\uparrow 0.019)$ |
| 640k | 50 | 0.2 | $36.58_{(0.67)}$ | $60.61_{(1.00)}$ | $83.30_{(1.18)}$ | $66.65_{(1.10)}$ | $34.53_{(0.47)}$ | $56.33(\downarrow 1.02)$ | $2.688(\uparrow 0.021)$ |
| 720k | 56.25 | 0.2 | $35.61_{(0.67)}$ | $60.94_{(1.00)}$ | $83.00_{(1.19)}$ | $67.14_{(1.10)}$ | $34.54_{(0.47)}$ | $56.25(\downarrow 1.10)$ | $2.687(\uparrow 0.020)$ |
| 800k | 62.5 | 0.2 | $35.20_{(0.67)}$ | $60.48_{(1.00)}$ | $83.40_{(1.20)}$ | $66.54_{(1.10)}$ | $34.45_{(0.47)}$ | $56.01(\downarrow 1.34)$ | $2.688(\uparrow 0.021)$ |

(b) 1B model, continually pre-trained. Note that $r = 0.4$ is early stopped due to its Pile validation loss increasing beyond the acceptable range.

| $N$ | $\rho\,(\%)$ | $r$ | Training tokens (B) | LAMBADA | ARC-E | Sciq | PIQA | HellaSwag | Avg. | Pile val. loss |
|---|---|---|---|---|---|---|---|---|---|---|
| | Pythia-1B-100k-ckpt | | | $55.66_{(0.69)}$ | $54.50_{(1.02)}$ | $83.00_{(1.19)}$ | $70.78_{(1.06)}$ | $36.97_{(0.48)}$ | 60.18 | 2.168 |
| 2.56M | 100 | 0.2 | 64 | $53.68_{(0.69)}$ | $51.47_{(1.03)}$ | $81.00_{(1.24)}$ | $68.77_{(1.08)}$ | $35.91_{(0.48)}$ | $58.17(\downarrow 2.01)$ | $2.184(\uparrow 0.016)$ |
| 2.56M | 100 | 0.4 | 24 | $52.38_{(0.70)}$ | $51.47_{(1.03)}$ | $80.70_{(1.25)}$ | $68.17_{(1.09)}$ | $34.95_{(0.48)}$ | $57.53(\downarrow 2.65)$ | $2.198(\uparrow 0.030)$ |
| 1.28M | 50 | 0.2 | 64 | $54.71_{(0.69)}$ | $52.86_{(1.02)}$ | $81.30_{(1.23)}$ | $68.99_{(1.08)}$ | $35.48_{(0.48)}$ | $58.67(\downarrow 1.51)$ | $2.189(\uparrow 0.022)$ |

Table 4: Detailed downstream performance for applying the random subsampling strategy to WikiBio. We use $\rho$ to denote the subsampling ratio. We report the accuracy (%) and standard deviation (%) in the format $\text{acc.}_{\text{(std. dev.)}}$ for each downstream task. Note that $r = 0.2$ is early stopped due to its Pile validation loss increasing beyond the acceptable range.

| $N$ | $\rho\,(\%)$ | $r$ | Training tokens (B) | LAMBADA | ARC-E | Sciq | PIQA | HellaSwag | Avg. | Pile val. loss |
|---|---|---|---|---|---|---|---|---|---|---|
| | Pythia-1B-100k-ckpt | | | $50.86_{(0.70)}$ | $52.10_{(1.03)}$ | $83.70_{(1.17)}$ | $67.14_{(1.10)}$ | $34.09_{(0.47)}$ | 57.58 | 2.255 |
| 277k | 100 | 0.1 | 32 | $50.77_{(0.70)}$ | $48.95_{(1.03)}$ | $80.80_{(1.25)}$ | $66.43_{(1.10)}$ | $33.16_{(0.47)}$ | $56.02(\downarrow 1.56)$ | $2.286(\uparrow 0.031)$ |
| 277k | 100 | 0.15 | 32 | $49.12_{(0.70)}$ | $49.66_{(1.03)}$ | $81.80_{(1.22)}$ | $66.38_{(1.10)}$ | $32.84_{(0.47)}$ | $55.96(\downarrow 1.62)$ | $2.292(\uparrow 0.037)$ |
| 277k | 100 | 0.2 | 20 | $49.37_{(0.70)}$ | $49.87_{(1.03)}$ | $79.70_{(1.27)}$ | $65.40_{(1.11)}$ | $33.08_{(0.47)}$ | $55.48(\downarrow 2.10)$ | $2.306(\uparrow 0.051)$ |
| 69k | 25 | 0.1 | 32 | $48.63_{(0.70)}$ | $50.59_{(1.03)}$ | $81.00_{(1.24)}$ | $66.49_{(1.10)}$ | $33.16_{(0.47)}$ | $55.97(\downarrow 1.54)$ | $2.286(\uparrow 0.031)$ |
| 137k | 50 | 0.1 | 32 | $50.30_{(0.70)}$ | $50.38_{(1.03)}$ | $78.80_{(1.29)}$ | $66.27_{(1.10)}$ | $33.16_{(0.47)}$ | $55.78(\downarrow 1.80)$ | $2.285(\uparrow 0.030)$ |
| 208k | 75 | 0.1 | 32 | $50.34_{(0.70)}$ | $49.20_{(1.03)}$ | $80.10_{(1.26)}$ | $66.97_{(1.10)}$ | $33.19_{(0.47)}$ | $55.96(\downarrow 1.62)$ | $2.286(\uparrow 0.031)$ |

Table 5: Detailed downstream performance for applying the compact knowledge mixing strategy on WikiBio. We use $\tau$ to denote the CKM ratio. We report the accuracy (%) and standard deviation (%) in the format $\text{acc.}_{\text{(std. dev.)}}$ for each downstream task. Note that $r = 0.2$ is early stopped due to its Pile validation loss increasing beyond the acceptable range.

| $r$ | $\tau\,(\%)$ | Training tokens (B) | LAMBADA | ARC-E | Sciq | PIQA | HellaSwag | Avg. | Pile val. loss |
|---|---|---|---|---|---|---|---|---|---|
| | Pythia-1B-100k-ckpt | | $50.86_{(0.70)}$ | $52.10_{(1.03)}$ | $83.70_{(1.17)}$ | $67.14_{(1.10)}$ | $34.09_{(0.47)}$ | 57.58 | 2.255 |
| 0.1 | 0 | 32 | $50.77_{(0.70)}$ | $48.95_{(1.03)}$ | $80.80_{(1.25)}$ | $66.43_{(1.10)}$ | $33.16_{(0.47)}$ | $56.02(\downarrow 1.56)$ | $2.286(\uparrow 0.031)$ |
| 0.15 | 0 | 32 | $49.12_{(0.70)}$ | $49.66_{(1.03)}$ | $81.80_{(1.22)}$ | $66.38_{(1.10)}$ | $32.84_{(0.47)}$ | $55.96(\downarrow 1.62)$ | $2.292(\uparrow 0.037)$ |
| 0.2 | 0 | 20 | $49.37_{(0.70)}$ | $49.87_{(1.03)}$ | $79.70_{(1.27)}$ | $65.40_{(1.11)}$ | $33.08_{(0.47)}$ | $55.48(\downarrow 2.10)$ | $2.306(\uparrow 0.051)$ |
| 0.1 | 10 | 32 | $49.70_{(0.70)}$ | $49.54_{(1.03)}$ | $80.40_{(1.26)}$ | $66.32_{(1.10)}$ | $33.11_{(0.47)}$ | $55.81(\downarrow 1.77)$ | $2.287(\uparrow 0.032)$ |
| 0.1 | 30 | 32 | $50.11_{(0.70)}$ | $49.12_{(1.03)}$ | $80.20_{(1.26)}$ | $66.54_{(1.10)}$ | $33.11_{(0.47)}$ | $55.82(\downarrow 1.76)$ | $2.285(\uparrow 0.030)$ |
| 0.1 | 60 | 32 | $49.99_{(0.70)}$ | $49.41_{(1.03)}$ | $80.00_{(1.27)}$ | $65.78_{(1.11)}$ | $32.99_{(0.47)}$ | $55.63(\downarrow 1.76)$ | $2.286(\uparrow 0.031)$ |

# E Experimental Details

## E.1 General Setup

**Code Base and Hyperparameters.** Our experiments use the GPT-NeoX library [Andonian et al., 2023]. For all experiments, we set the batch size as 512 and the sequence length as 2048. For all the experiments in Section 3, we use a Warmup-Stable-Decay (WSD) learning rate schedule with a peak learning rate of $10^{-3}$. We allocate 160 steps for warmup and the final 10% steps for cooldown. We keep other hyperparameters consistent with those used in Pythia.

**Hardware.** We train models of sizes 70M and 160M using 8 NVIDIA RTX 6000 Ada GPUs, while models of sizes 410M and 1B are trained using either 16 NVIDIA RTX 6000 Ada GPUs or 8 NVIDIA A100 GPUs. The estimated runtime required to train each model size on 1B tokens is detailed in Table 6. Consequently, a typical run training a 410M model on 32B tokens takes approximately 32 hours, whereas the longest run, which trains a 1B model on 64B tokens, exceeds five days.

Table 6: Estimated runtime required to train each model size on 1B tokens on our hardware.

| Model size | Hardware | Runtime (h) per billion tokens. |
|---|---|---|
| 70M | 8xNVIDIA RTX 6000 Ada | 0.25 |
| 160M | | 0.70 |
| 410M | 16xNVIDIA RTX 6000 Ada | 1.0 |
| 1B | or 8xNVIDIA A100 | 2.0 |
| 2.8B | 8xNVIDIA A100 | 5.8 |
| 6.9B | | 16.89 |

**Implementation of Data Mixing.** Let $S$ denote the total number of training tokens. Then, the model sees $rS$ tokens from the knowledge-dense dataset and $(1-r)S$ tokens from the web data. Let $S_1$ and $S_2$ denote the total sizes (in tokens) of the knowledge-dense dataset and web data. Since $S_1$ is small ($< 1B$ tokens) and $S_2$ is large ($> 1T$ tokens), for the training horizons $S$ considered in our experiments, we typically have $rS > S_1$ and $(1-r)S < S_2$. In this case, we replicate the knowledge-dense dataset $rS/S_1$ times, sample a random $(1-r)S$-token subset from the web data, and then shuffle the combined data.

**Licenses for the Public Assets.** The FineWeb-Edu and OpenWebMath datasets are under the ODC-BY License. The Pile dataset is under the MIT License. The Pythia model suite and the gpt-neox-library are under the Apache License 2.0. All of them are open for academic usage.

### E.2 Details of Dataset Construction

#### E.2.1 Constructing the SynBio Dataset

To generate names, we collect a list of 400 common first names, 400 common middle names, and 1000 common last names, resulting in $1.6 \times 10^8$ unique names. To generate SynBio-$N$, we sample $N$ names from this set without replacement. For each individual, the value for each attribute is randomly assigned as follows: birth date (1–28 days × 12 months × 100 years spanning 1900–2099), birth city (from 200 U.S. cities), university (from 300 institutions), major (from 100 fields of study), and employer (from 263 companies). Each attribute is paired with five sentence templates, which are used to convert (name, attribute, value) triplets into natural text descriptions. A complete list of sentence templates is provided in Table 7, and an example of a synthetic biography can be found in Table 8.

Table 7: Sentence templates to generate the SynBio Dataset.

| Attribute | Template |
|---|---|
| Birth date | {name} was born on {birth date}.
{name} came into this world on {birth date}.
{name}'s birth date is {birth date}.
{name}'s date of birth is {birth date}.
{name} celebrates {possessive pronoun} birthday on {birth date}. |
| Birth city | {name} spent {possessive pronoun} early years in {birth city}.
{name} was brought up in {birth city}.
{name}'s birthplace is {birth city}.
{name} originates from {birth city}.
{name} was born in {birth city}. |
| University | {name} received mentorship and guidance from faculty members at {university}.
{name} graduated from {university}.
{name} spent {possessive pronoun} college years at {university}.
{name} completed {possessive pronoun} degree at {university}.
{name} completed {possessive pronoun} academic journey at {university}. |
| Major | {name} completed {possessive pronoun} education with a focus on {major}.
{name} devoted {possessive pronoun} academic focus to {major}.
{name} has a degree in {major}.
{name} focused {possessive pronoun} academic pursuits on {major}.
{name} specialized in the field of {major}. |
| Employer | {name} is employed at {employer}.
{name} a staff member at {employer}.
{name} is associated with {employer}.
{name} is engaged in work at {employer}.
{name} is part of the team at {employer}. |

Table 8: An example of a synthetic biography. The values that we expect the model to recall during evaluation are underlined.

Gracie Tessa Howell's birth date is August 09, 1992. Gracie Tessa Howell's birthplace is St. Louis, MO. Gracie Tessa Howell received mentorship and guidance from faculty members at Santa Clara University. Gracie Tessa Howell has a degree in Robotics. Gracie Tessa Howell is engaged in work at Truist Financial.

### E.2.2 Constructing the WikiBio Dataset

To create the WikiBio dataset, we first query Wikidata to gather names and birth dates of individuals from 16 common occupations. We then identify each person's Wikipedia page by matching their name with the page title. We retain the first paragraph of each page, as it typically provides a short summary of the person's life and contains key biographical information. The detailed composition is listed in Table 9. Finally, to align with the evaluation setup in Section 6, we filter the dataset to ensure both the person's occupation and birth date are explicitly mentioned.

Inspired by Allen-Zhu and Li [2023], we employ Llama-3.1-70B-Instruct to paraphrase each biography ten times, thereby simulating the real-world scenario where models encounter different variations of the same person's information during training. See Table 10 for the prompt for paraphrasing. An example of the original text and the paraphrased versions are presented below.

- **Original text**: Rebecca Jo Budig (born June 26, 1973) is an American actress and television presenter. Her career began in 1993, and in 1995, she was cast in the role of Michelle Bauer on the CBS soap opera Guiding Light. In 1999, she was cast as Greenlee Smythe on the ABC soap opera All My Children; she held the role off-and-on until the network series finale in 2011. In 2015, she was cast in the role of Hayden Barnes on General Hospital. In 2019, Budig was cast on L.A.'s Finest, as Carlene Hart, the drug trafficking-soccer mom of two young school-age children, a boy and a girl.

- **Paraphrased versions**:

  1. Born on June 26, 1973, Rebecca Jo Budig is an American television presenter and actress with a career spanning nearly three decades. Her professional journey began in 1993 and has included notable roles such as Michelle Bauer on Guiding Light and Greenlee Smythe on All My Children. After playing the latter role on-and-off until 2011, she went on to portray Hayden Barnes in General Hospital in 2015 and Carlene Hart in the 2019 television series L.A.'s Finest.

  2. With a diverse career in television, Rebecca Jo Budig, born June 26, 1973, has established herself as a talented actress and presenter. Her career milestones include her roles as Michelle Bauer in the CBS soap opera Guiding Light, and Greenlee Smythe in All My Children. Her portrayal of Greenlee spanned several years, concluding with the show's finale in 2011. Budig continued to expand her repertoire with roles in General Hospital and L.A.'s Finest.

  3. Rebecca Jo Budig is a versatile American actress and television host, born on June 26, 1973. Since her career began in 1993, she has landed prominent roles in several television series. One of her earliest notable roles was Michelle Bauer in Guiding Light, followed by her portrayal of Greenlee Smythe in All My Children, a character she played until the series' conclusion in 2011. In the years that followed, she appeared in General Hospital and L.A.'s Finest.

  4. Rebecca Jo Budig, an American actress and television presenter, was born on June 26, 1973. She began her career two decades later, securing the role of Michelle Bauer on Guiding Light. Budig's subsequent roles have included Greenlee Smythe on All My Children, a part she played intermittently until the series ended in 2011. Her later appearances include a role in General Hospital and as Carlene Hart in the series L.A.'s Finest.

  5. American actress Rebecca Jo Budig was born on June 26, 1973. Her television career, which began in 1993, encompasses multiple notable roles, such as Michelle Bauer on the soap opera Guiding Light and Greenlee Smythe on All My Children. She portrayed the latter character until the series finale in 2011. Budig later appeared as Hayden Barnes in General Hospital and took on the role of Carlene Hart in L.A.'s Finest.

  6. Since launching her career in 1993, Rebecca Jo Budig has established herself as a talented actress and television presenter in the United States. Born on June 26, 1973, she has appeared in a range of notable roles, including Michelle Bauer on Guiding Light and Greenlee Smythe on All My Children. The latter role spanned several years, concluding with the show's finale in 2011. Her subsequent appearances include General Hospital and L.A.'s Finest.

  7. Rebecca Jo Budig, born on June 26, 1973, has enjoyed a successful career in American television as an actress and presenter. Her breakout role came in 1995 when she was cast as Michelle Bauer on Guiding Light. Later, she played the character Greenlee Smythe on All My Children, a part she held intermittently until the show's finale in 2011. Her more recent roles include appearances in General Hospital and as Carlene Hart in L.A.'s Finest.

8. Born on June 26, 1973, Rebecca Jo Budig is a talented American actress and television presenter. Since her career began in 1993, she has appeared in various television series. Notable roles include her portrayal of Michelle Bauer on the soap opera Guiding Light, as well as Greenlee Smythe on All My Children. Budig continued to expand her acting repertoire with roles in General Hospital and L.A.'s Finest, including her portrayal of Carlene Hart.

9. As an American actress and television host, Rebecca Jo Budig has had a diverse career spanning nearly three decades. Born on June 26, 1973, she began her professional journey in 1993. Her notable roles include Michelle Bauer on Guiding Light and Greenlee Smythe on All My Children, a character she played until the series finale in 2011. Her subsequent appearances include General Hospital and the series L.A.'s Finest, where she portrayed Carlene Hart.

10. With a career in television that began in 1993, Rebecca Jo Budig, born June 26, 1973, has established herself as a versatile actress and presenter. Her early roles include Michelle Bauer on Guiding Light, while her breakout role came as Greenlee Smythe on All My Children. She continued to portray Greenlee intermittently until the show's finale in 2011. Her later roles include appearances in General Hospital and L.A.'s Finest, where she took on the role of Carlene Hart.

Table 9: Detailed Composition of WikiBio.

| Occupation | Num. Wikipedia biographies |
| --- | --- |
| Singer | 18,482 |
| Actor | 31,846 |
| Politician | 38,653 |
| Businessperson | 8,068 |
| Mathematician | 5,093 |
| Physicist | 4,296 |
| Writer | 26,746 |
| Football player | 56,547 |
| Basketball player | 16,956 |
| Sport shooter | 3,156 |
| Tennis plater | 7,602 |
| Swimmer | 9,108 |
| Painter | 12,927 |
| Volleyball player | 3,556 |
| Composer | 13,719 |
| Athlete | 18,013 |
| Total | 274,768 |

Table 10: The prompt for paraphrasing the first paragraph of Wikipedia documents.

```
I am creating the training data for an LLM. I
↪ would like to teach it to flexibly extract
↪ knowledge from a Wikipedia paragraph.
↪ Therefore, I want to diversify the Wikipedia
↪ paragraphs as much as possible so that the
↪ model can learn the actual relationships
↪ between entities, rather than just memorizing
↪  the text. Please assist with the
↪ paraphrasing task. Paraphrase the following
↪ Wikipedia paragraph about {Wikipedia document
   title} 10 times. Aim to make the paraphrased
↪ versions as varied as possible. Ensure all
↪ essential information is retained,
↪ particularly the information about the
↪ birthday and the occupation.
```

### E.3 Constructing the SlopeQA Dataset

Every time the model sees a slope calculation example, we first uniformly sample $x_1, y_1, x_2, y_2$ from $\{0, 1, \cdots, 99\}$ (ensuring $x_1 \neq x_2$), and then apply randomly chose question and step-by-step answer templates. We prompt GPT-4o to generate diverse question and answer templates, as shown in Tables 11 and 12. The final answer is expressed as the simplified fraction.

Table 11: Question templates for the slope calculation subtask.

---

```
Q: Find the slope of the line passing through ({x1},{y1}) and (
↪ {x2},{y2}).

Think step-by-step.
```

---

```
Q: What is the slope of the line passing through ({x1},{y1}) and
↪ ({x2},{y2})?

Think step-by-step.
```

---

```
Q: Compute the slope of the line going through ({x1},{y1}) and (
↪ {x2},{y2}).

Think step-by-step.
```

---

```
Q: Determine the slope of the line connecting ({x1},{y1}) and (
↪ {x2},{y2}).

Think step-by-step.
```

---

```
Q: If a line goes through ({x1},{y1}) and ({x2},{y2}), what is its
↪ slope?

Think step-by-step.
```

---

Table 12: Answer templates for the slope calculation subtask.

---

```
A: Recall the slope formula:
   k=(y2-y1)/(x2-x1)

1. Let (x1,y1)=({x1},{y1}) and (x2,y2)=({x2},{y2}).

2. Compute the difference:
   x2-x1={x2}-{x1}={x2-x1}
   y2-y1={y2}-{y1}={y2-y1}

3. Plug into the formula:
   k={x2-x1}/{y2-y1}

### Final Answer:
   k={k}
```

---

```
A:
1. Identify the coordinates:
   First point: (x1,y1)=({x1},{y1})
   Second point: (x2,y2)=({x2},{y2})

2. Compute the difference in y-values:
   {y2}-{y1}={y2-y1}

3. Compute the difference in x-values:
   {x2}-{x1}={x2-x1}

4. Divide the differences:
   k={y2-y1}/{x2-x1}

### Final Answer:
   k={k}
```

---

```
A: We use the slope formula:
   k=(y2-y1)/(x2-x1)

   1. Plug in the coordinates:
   k=({y2}-{y1})/({x2}-{x1})

   2. Simplify the numerator and denominator:
   k={y2-y1}/{x2-x1}

   3. Simplify the fraction:
   k={k}

### Final Answer:
   k={k}
```

---

```
A: To find the slope, we use the slope formula:
   k=(y2-y1)/(x2-x1)

   1. We are given two points:({x1},{y1}),({x2},{y2})

   2. Plug the values into the formula:
      k=({y2}-{y1})/({x2}-{x1})={y2-y1}/{x2-x1}

### Final Answer:
   k={k}
```

---

```
A: The slope k between (x1,x2) and (y1,y2) is given by:
k=(y2-y1)/(x2-x1)

1. From the problem: (x1,y1)=({x1},{y1}),(x2,y2)=({x2},{y2})

2. Substituting into the formula:
   k=({y2}-{y1})/({x2}-{x1})={y2-y1}/{x2-x1}

3. Simplify:
   k={y2-y1}/{x2-x1}

### Final Answer:
   k={k}
```

## E.4 Constructing the Max-over-N Dataset

We design a new task named "Max-over-N", where the model is tasked with outputting the maximum number of a list of $N$ integers, each randomly sampled from $\{0, 1, \cdots, 99\}$. In our experiments, we set $N = 30$.

Table 13: An example of the Max-over-N subtask.

```
Q: Find the maximum value of the following list:
[47, 83, 38, ...]
Think step-by-step.

A:
Compare 47 and 83. Keep 83.
Compare 83 and 38. Keep 83.
```

## E.5 Details of the Fitting Process

We use $T$ to denote the required training steps to reach 40% accuracy and $r$ to denote the mixing ratio.

**Fitting the exponential function.** We fit $T$ with respect to $r$ for all $r \geq 0.3$ using the function $T(r) = A \exp(B/r)$, where $A$ and $B$ are coefficients to be fitted. Taking logarithmic on both sides, we obtain a linear function $\log T = \log A + B/r$. By fitting $\log T$ against $1/r$ with linear regression, we obtain $\log A \approx -0.25512, B \approx 1.5137$ with goodness-of-fit $R^2 = 0.9980$.

**Fitting the power-law function.** We fit $T$ with respect to $r$ for all $r \in \{0.3, 0.4, 0.45, 0.5, 0.55\}$ using the function $T(r) = Cr^{-D}$, where $C$ and $D$ are coefficients to be fitted. Taking logarithmic on both sides, we obtain a linear function $\log T = \log C - D \log r$. By fitting $\log T$ against $\log r$ with linear regression, we obtain $C \approx 0.098158, D \approx 3.83878$ with goodness-of-fit $R^2 = 0.9853$.

## E.6 Details of Estimating the Threshold Popularity

Following Mallen et al. [2023], we evaluate models using 15-shot prompting. We use the prompt presented in Table 14 for evaluatioin and allow models to generate up to 128 tokens with greedy decoding. To assess answer correctness, we employ Llama-3.1-8B-Instruct as a judge. Specifically, we instruct the Llama-3.1-8B-Instruct model to evaluate the semantic similarity between the model-generated answer and the reference answer provided in PopQA. The prompt used for the Llama judge is detailed in Table 15.

After judging the correctness of each answer, we use Algorithm 1 to estimate the popularity threshold. In our experiments, we set the target accuracy $\alpha_{\text{target}} = 60\%$ and the fault tolerance level $N_{\text{fail}} = 5$.

Table 14: The prompt for evaluating models on PopQA.

```
You are a helpful assistant. I want to test your knowledge level.
↪ Here are a few examples.

{few shot examples text with templates}

Now, I have a question for you. Please respond in just a few words
↪ , following the style of the examples provided above.
```

Table 15: The prompt for testing synonym.

```
<|begin_of_text|><|start_header_id|>system<|
↪ end_header_id|>
Cutting Knowledge Date: December 2023
Today Date: 19 Dec 2024

You are a linguistic expert specializing in synonyms.
↪ Your task is to determine whether two given English
↪ words are synonyms or not. A synonym is a word that
↪ has a very similar meaning to another word and can
↪ often replace it in sentences without significantly
↪ changing the meaning.

For each pair of words provided:
1. Analyze their meanings and typical usage.
2. Decide whether they are synonyms (Yes/No).
3. Provide a brief explanation for your decision.

Here are some examples to guide you:

Words: "happy" and "joyful"
Yes
Explanation: Both words describe a state of being
↪ pleased or content and are often interchangeable in
↪ most contexts.

Words: "run" and "jog"
No
Explanation: While both refer to forms of movement, "run"
↪  typically implies a faster pace than "jog."

Words: "angry" and "frustrated"
No
Explanation: Although both express negative emotions, "
↪ angry" implies strong displeasure or rage, while "
↪ frustrated" conveys annoyance due to obstacles or
↪ failure.

<|eot_id|><|start_header_id|>user<|end_header_id|>

Words: {} and {}
<|eot_id|><|start_header_id|>assistant<|end_header_id|>
```

**Algorithm 1:** Estimate Threshold Popularity

---

1: **Input:**
2:    - $x$: A list of popularity values for each data point, where $x_i$ represents the popularity of the $i$-th data point.
3:    - $y$: A list of binary values indicating the correctness of the model's response, where $y_i = 1$ if the model answers the $i$-the question correctly, and $y_i = 0$ otherwise.
4:    - $\alpha_{\text{target}}$: The target accuracy.
5:    - $N_{\text{fail}}$: The maximum number of failures before termination, denoting the fault tolerance level.
6: **Output:**
7:    - $P_{\text{thres}}$: The threshold popularity.
8: Initialize correct count: sum_correct $\leftarrow 0$
9: Initialize error count: $e \leftarrow 0$
10: Sort $(x, y)$ by $x$ in ascending order and store the indices in a list $I$.
11: Initialize loop variable $j \leftarrow \text{len}(x) - 1$
12: Initialize flag counter flag $\leftarrow 0$
13: **while** $j \geq 0$ **do**
14:    $k \leftarrow j$
15:    **while** $k \geq 0$ **and** $x_{I_k} = x_{I_j}$ **do**
16:      $k \leftarrow k - 1$
17:    **end while**
18:    **for** $l = k + 1$ **to** $j$ **do**
19:      $i \leftarrow I_l$
20:      sum_correct $\leftarrow$ sum_correct $+ y_i$
21:    **end for**
22:    **if** $\frac{\text{sum\_correct}}{\text{len}(x) - k - 1} <$ set_threshold **then**
23:      $e \leftarrow e + 1$
24:    **end if**
25:    **if** $e = N_{\text{fail}}$ **then**
26:      **Return:** $x_{I_j}$ {Return the threshold popularity}
27:    **end if**
28:    $j \leftarrow k$
29: **end while**
30: **Return:** $x_{I_0}$ {If no such point is found, return the smallest popularity value}

---

### E.7 Experimental Details for Strategies to Enhance Knowledge Acquisition

This subsection presents the experimental details for Section 6.

**Evaluation Details for WikiBio.** For simplicity, we focus on how well the model memorizes one specific type of fact: the birth date of a person, which is ensured to be mentioned in WikiBio. Specifically, for each fact, which can be represented as a (name, occupation, birth date) triplet, we prompt the model with "The {occupation} {name} was born on" and consider the response correct if it contains the correct birth year and month. The occupation is included in the evaluation prompt not only to avoid prompts that are exactly identical to the training data but also to provide additional context for disambiguation.

**Implementation Details of CKM.** When we apply CKM to WikiBio, we augment the dataset by adding compact tuple representations. To maintain the same token budget for WikiBio after augmentation (as $r$ and the total training tokens are fixed), we proportionally reduce the number of epochs. Although the model completes fewer epochs over the dataset, each fact's frequency per epoch is increased, boosting its total exposure during training. For example, in Figure 8(b), setting $\tau = 0.1, 0.3$ and $0.6$ correspond to roughly 2x, 3x, and 4x increases in fact frequency, respectively. Each time models encounter the tuple-form data point, the order of birth date and occupation is randomly flipped.

**Experimental Details of Random Subsampling.** In Figure 8(a), we train all models from scratch on the mixture of FineWeb-Edu and SynBio-1.28M using the cosine learning rate schedule with a peak value of $10^{-3}$. In Figures 8(b) and 8(c), following Zhu et al. [2024], we continually pre-train

intermediate checkpoints of Pythia models. This strategy allows us to use a larger learning rate without experiencing extreme loss spikes. Specifically, we continually pre-train 410M and 1B Pythia models from their respective 100k-step checkpoint with a constant learning rate of $8.7 \times 10^{-5}$, which corresponds to the original learning rate used at step 100k in the Pythia model training.

# F   Proofs of Theoretical Results

We follow the notations in Section 4. We use $H(\cdot)$ to denote the entropy and $I(\cdot\,;\cdot)$ to denote the mutual information.

We define a data distribution $\mathcal{D}$ as a distribution over $(x, y)$, where $x$ is an input and $y$ is a token. A data universe $\mathcal{U} = (\mathcal{P}, \mathcal{D}_\theta)$ is defined by a prior $\mathcal{P}$ over a latent variable $\theta$ and a family of data distributions $\mathcal{D}_\theta$ indexed by $\theta$.

A predictor $h$ is a function that maps $x$ to a distribution over $y$. A learning algorithm $\mathcal{A}$ is a procedure that takes samples from a data distribution $\mathcal{D}$ of $(x, y)$ and outputs a predictor $h \sim \mathcal{A}(\mathcal{D})$ in the end. For a given predictor $h$, we measure its performance by the expected cross-entropy loss

$$\mathcal{L}(h; \mathcal{D}) := \mathbb{E}_{(x,y)\sim\mathcal{D}}[-\log p(y \mid h, x)], \tag{7}$$

where $p(y \mid h, x)$ denotes the predicted distribution of $y$ given $x$ by the predictor $h$, and $\log$ is in base 2 for convenience. For a data universe $\mathcal{U} = (\mathcal{P}, \mathcal{D}_\theta)$, we measure the performance of a learning algorithm $\mathcal{A}$ by its expected loss over all data distributions $\mathcal{D}_\theta$ with respect to the prior $\mathcal{P}$:

$$\bar{\mathcal{L}}_\mathcal{P}(\mathcal{A}) := \mathbb{E}_{\theta\sim\mathcal{P}}\mathbb{E}_{h\sim\mathcal{A}(\mathcal{D}_\theta)}[\mathcal{L}(h; \mathcal{D}_\theta)]. \tag{8}$$

We use the mutual information $I(\mathcal{A}(\mathcal{D}_\theta); \mathcal{D}_\theta)$ as a measure of the *effective model capacity* for the predictor picked by $\mathcal{A}$ on $\mathcal{D}_\theta$, where $\theta$ is sampled from $\mathcal{Q}$.

Same as Definition 4.1, for a data universe $\mathcal{U} = (\mathcal{P}, \mathcal{D}_\theta)$ and $M > 0$, we define the best achievable loss under the capacity constraint $M$ as

$$F_\mathcal{P}(M) := \inf_\mathcal{A} \left\{ \bar{\mathcal{L}}_\mathcal{P}(\mathcal{A}) : I(\mathcal{A}(\mathcal{D}_\theta); \mathcal{D}_\theta) \le M \right\}, \tag{9}$$

where the infimum is taken over all learning algorithms. An optimal $M$-bounded-capacity learner is a learning algorithm $\mathcal{A}$ such that $I(\mathcal{A}(\mathcal{D}_\theta); \mathcal{D}_\theta) \le M$ and $\bar{\mathcal{L}}_\mathcal{P}(\mathcal{A}) = F_\mathcal{P}(M)$.

## F.1   Convexity of the Best Achievable Loss

It is easy to see that $F_\mathcal{P}(M)$ is non-negative and non-increasing in $M$. A classic result in rate-distortion theory is that the rate-distortion function is convex. This further implies that $F_\mathcal{P}(M)$ is convex in $M$. Here we present it as a lemma for completeness.

**Lemma F.1.** *For any data universe $\mathcal{U} = (\mathcal{P}, \mathcal{D}_\theta)$, $F_\mathcal{P}(M)$ is convex in $M$.*

*Proof.* Let $\epsilon > 0$ be any positive number. Let $\mathcal{A}_1$ be a learning algorithm that achieves a loss $\le F_\mathcal{P}(M_1) + \epsilon$ with mutual information $I_1(\mathcal{A}(\mathcal{D}_\theta); \mathcal{D}_\theta) \le M_1$ and $\mathcal{A}_2$ be a learning algorithm that achieves a loss $F_\mathcal{P}(M_2) + \epsilon$ with mutual information $I_2(\mathcal{A}(\mathcal{D}_\theta); \mathcal{D}_\theta) \le M_2$.

Let $\mathcal{A}$ be a new learning algorithm that outputs the same as $\mathcal{A}_1$ with probability $1 - p$ and the same as $\mathcal{A}_2$ with probability $p$. Then the mutual information between $\mathcal{A}(\mathcal{D}_\theta)$ and $\mathcal{D}_\theta$ is

$$I(\mathcal{A}(\mathcal{D}_\theta); \mathcal{D}_\theta) = (1 - p)I(\mathcal{A}_1(\mathcal{D}_\theta); \mathcal{D}_\theta) + pI(\mathcal{A}_2(\mathcal{D}_\theta); \mathcal{D}_\theta)$$
$$\le (1 - p)M_1 + pM_2.$$

By linearity of expectation, the expected loss of $\mathcal{A}$ can be bounded as

$$\mathbb{E}_{\theta\sim\mathcal{P}(\theta)}[\mathcal{L}(\mathcal{A}(\mathcal{D}_\theta); \mathcal{D}_\theta)] = (1 - p)\mathbb{E}_{\theta\sim\mathcal{P}(\theta)}[\mathcal{L}(\mathcal{A}_1(\mathcal{D}_\theta); \mathcal{D}_\theta)] + p\mathbb{E}_{\theta\sim\mathcal{P}(\theta)}[\mathcal{L}(\mathcal{A}_2(\mathcal{D}_\theta); \mathcal{D}_\theta)]$$
$$\le (1 - p)F_\mathcal{P}(M_1) + pF_\mathcal{P}(M_2) + 2\epsilon.$$

Therefore, we have

$$F_\mathcal{P}((1 - p)M_1 + pM_2) \le \mathbb{E}_{\theta\sim\mathcal{P}(\theta)}[\mathcal{L}(\mathcal{A}(\mathcal{D}_\theta); \mathcal{D}_\theta)] \le (1 - p)F_\mathcal{P}(M_1) + pF_\mathcal{P}(M_2) + 2\epsilon,$$

taking $\epsilon \to 0$ finishes the proof. □

## F.2 Proofs for the Warmup Case

**Definition F.2** (Factual Data Universe). We define a fact as a pair $(X, y)$, where $X$ is a set of inputs and $y$ is a target token. A factual data universe is a data universe $\mathcal{U} = (\mathcal{P}, \mathcal{D}_\theta)$ containing $K$ random facts $(X_1, y_1), \ldots, (X_K, y_K)$ in the following way:

1. $X_1, \ldots, X_K$ are $K$ disjoint sets of inputs, and $y_1, \ldots, y_K$ are random tokens;

2. $\theta$ is structured as $(y_1, \ldots, y_K)$. Given $\theta = (y_1, \ldots, y_K)$, the data distribution $\mathcal{D}_\theta$ satisfies that for all $x \in X_i$, $\mathcal{D}_\theta(y \mid x_i)$ is a point mass at $y_i$;

3. For all $\theta$, the input distribution $\mathcal{D}_\theta(x)$ is the same;

4. For all $\theta$, the target distribution $\mathcal{D}_\theta(y \mid x)$ is the same for all $x \notin \bigcup_{i=1}^K X_i$;

5. The prior distribution $\mathcal{P}$ over $\theta$ is given by the product distribution $\mathcal{P}(y_1, y_2, \ldots, y_K) = \prod_{k=1}^K \mathcal{Y}_k(y_k)$, where $\mathcal{Y}_k$ is a fixed prior distribution over $y_k$.

The exposure frequency of each random fact is defined as the total probability that an input $x \in X_i$ occurs in $\mathcal{D}_\theta$.

**Theorem F.3** (Theorem 4.2, restated). *For a factual data universe $\mathcal{U} = (\mathcal{P}, \mathcal{D}_\theta)$ with $K$ random facts, if all the facts have the same exposure frequency p, then*

$$F_{\mathcal{P}}(M) = C + p \cdot \max\{H_{\mathrm{tot}} - M, 0\}, \tag{10}$$

*where $H_{\mathrm{tot}} := \sum_{i=1}^K H(\mathcal{Y}_i)$ and $C := F_{\mathcal{P}}(\infty)$.*

*Proof.* First, we prove a lower bound for $F_{\mathcal{P}}(M)$. For any learning algorithm $\mathcal{A}$ with $I(\mathcal{A}(\mathcal{D}_\theta); \mathcal{D}_\theta) \leq M$,

$$\begin{aligned}
\bar{\mathcal{L}}_{\mathcal{P}}(\mathcal{A}(\mathcal{D}_\theta)) &= \mathbb{E}_{\theta \sim \mathcal{P}} \mathbb{E}_{(x,y) \sim \mathcal{D}_\theta} \mathbb{E}_{h \sim \mathcal{A}(\mathcal{D}_\theta)} [-\log p(y \mid h, x)] \\
&= \mathbb{E}_x \mathbb{E}_{\theta \sim \mathcal{P}} \mathbb{E}_{y \sim \mathcal{D}_\theta(\cdot \mid x)} \mathbb{E}_{h \sim \mathcal{A}(\mathcal{D}_\theta)} \mathbb{E}[-\log p(y \mid h, x)] \\
&\geq \underbrace{\mathbb{E}_x \left[ \mathbb{1}_{\{x \in \bigcup_{i=1}^K X_i\}} H_{\theta \sim \mathcal{P}}(\mathcal{D}_\theta(\cdot \mid x)) \right]}_{=:C_0} + p \left[ \sum_{i=1}^K (H(\mathcal{Y}_i) - I(\mathcal{A}(\mathcal{D}_\theta); y_i)) \right]_+ \\
&\geq C_0 + p \left[ H_{\mathrm{tot}} - I(\mathcal{A}(\mathcal{D}_\theta); \mathcal{D}_\theta) \right]_+ \\
&\geq C_0 + p \left[ H_{\mathrm{tot}} - M \right]_+.
\end{aligned}$$

For upper bounds, we first show that $F_{\mathcal{P}}(M) \leq C_0$ for all $M \geq H_{\mathrm{tot}}$. Let $\mathcal{A}_1$ be the learning algorithm that inputs $\mathcal{D}_\theta$ and outputs the predictor $h$ that always outputs the token $y_i$ for the input $x \in X_i$. For all the other inputs $x$, the predictor just outputs $h(y \mid h, x) = \mathbb{E}_{\theta \sim \mathcal{P}}[\mathcal{D}_\theta(y \mid x)]$. Both $\mathcal{A}_1(\mathcal{D}_\theta)$ and $\mathcal{D}_\theta$ can be transformed from $\theta$ with a reversible function, so

$$I(\mathcal{A}_1(\mathcal{D}_\theta); \mathcal{D}_\theta) = H(\theta) = \sum_{i=1}^K H(\mathcal{Y}_i) = H_{\mathrm{tot}}.$$

It is easy to see that $\bar{\mathcal{L}}_{\mathcal{P}}(\mathcal{A}_1) = C_0$. This implies that $F_{\mathcal{P}}(M) \leq C_0$ for all $M \geq H_{\mathrm{tot}}$.

Now, if $M < H_{\mathrm{tot}}$, we construct a learning algorithm $\mathcal{A}_q$ that outputs the same as $\mathcal{A}_1$ with probability $q$ and outputs $h(y \mid h, x) = \mathbb{E}_{\theta \sim \mathcal{P}}[\mathcal{D}_\theta(y \mid x)]$ with probability $1 - q$. Setting $q = \frac{M}{H_{\mathrm{tot}}}$, we have

$$I(\mathcal{A}_q(\mathcal{D}_\theta); \mathcal{D}_\theta) = q \cdot H_{\mathrm{tot}} = M.$$

By linearity of expectation, we also have $\bar{\mathcal{L}}_{\mathcal{P}}(\mathcal{A}_q(\mathcal{D}_\theta)) = \bar{\mathcal{L}}_{\mathcal{P}}(\mathcal{A}_1) + (1 - q) \cdot p \sum_{i=1}^K H(\mathcal{Y}_i)$. This implies that $F_{\mathcal{P}}(M) \leq \bar{\mathcal{L}}_{\mathcal{P}}(\mathcal{A}_1) + p \cdot \max\{H_{\mathrm{tot}} - M, 0\}$ for all $M < H_{\mathrm{tot}}$.

Putting all the pieces together finishes the proof. $\qquad\square$

### F.3 Proofs for the Data Mixing Case

**Definition F.4** (Mixture of Data Universes). Let $\mathcal{U}_1 = (\mathcal{P}_1, \mathcal{D}_{\theta_1}^{(1)})$ and $\mathcal{U}_2 = (\mathcal{P}_2, \mathcal{D}_{\theta_2}^{(2)})$ be two data universes. We mix them together to form a new data universe $\mathcal{U} = (\mathcal{P}, \mathcal{D}_\theta)$:

1. $\theta$ is structured as $(\theta_1, \theta_2)$. Given $\theta = (\theta_1, \theta_2)$, the data distribution $\mathcal{D}_\theta$ is formed as $\mathcal{D}_\theta = r\mathcal{D}_{\theta_1}^{(1)} + (1-r)\mathcal{D}_{\theta_2}^{(2)}$, where $r$ is called *the mixing ratio*;

2. The prior distribution $\mathcal{P}$ over $\theta$ is a joint distribution of $\mathcal{P}_1$ and $\mathcal{P}_2$.

In reality, mixing two datasets can be seen as mixing two data universes first and then sampling a data distribution from the mixed data universe. Here we consider the simplified case where the two data universes are so different from each other that they convey orthogonal information.

**Definition F.5** (Orthogonal Mixture of Data Universes). We say that $\mathcal{U}$ is an orthogonal mixture of $\mathcal{U}_1$ and $\mathcal{U}_2$ if

1. For any $x$ that is in both supports of $\mathcal{D}_{\theta_1}^{(1)}$ and $\mathcal{D}_{\theta_2}^{(2)}$, we have $\mathcal{D}_{\theta_1}^{(1)}(y \mid x) = \mathcal{D}_{\theta_2}^{(2)}(y \mid x)$ for all $\theta_1$ and $\theta_2$. In other words, the conditional distribution of the next token $y$ given the context $x$ remains consistent across both domains and is unaffected by variations in values of $\theta_1$ and $\theta_2$.

2. $\mathcal{P}(\theta_1, \theta_2) = \mathcal{P}_1(\theta_1) \cdot \mathcal{P}_2(\theta_2)$, i.e., $\theta_1$ and $\theta_2$ are independent.

Below, we first establish two lemmas that provide conditions for when the loss on the first domain will be very low or very high for an optimal $M$-bounded-capacity learner given an orthogonal mixture of two data universes. Then, we use these lemmas to prove Theorem 4.3.

We use $\mathrm{D}^-F(t)$ and $\mathrm{D}^+F(t)$ to denote the left and right derivatives of a function $F$ at a point $t$, respectively.

**Lemma F.6.** *Let $\mathcal{U} = (\mathcal{P}, \mathcal{D}_\theta)$ be an orthogonal mixture of $\mathcal{U}_1 = (\mathcal{P}_1, \mathcal{D}_{\theta_1}^{(1)})$ and $\mathcal{U}_2 = (\mathcal{P}_2, \mathcal{D}_{\theta_2}^{(2)})$ with mixing ratio $r$. For all $r \in (0, 1)$ and $M \geq 0$, if the following inequality holds,*

$$\frac{r}{1-r} < \frac{\mathrm{D}^-F_{\mathcal{P}_2}(M)}{\mathrm{D}^+F_{\mathcal{P}_1}(0)}, \tag{11}$$

*then for any optimal $M$-bounded-capacity learner $\mathcal{A}$ on $\mathcal{U}$, $\mathbb{E}_{\theta \sim \mathcal{P}}\mathbb{E}_{h \sim \mathcal{A}(\mathcal{D}_\theta)}[\mathcal{L}(h; \mathcal{D}_{\theta_1}^{(1)})] = F_{\mathcal{P}_1}(0)$.*

**Intuitive Explanation.** We define $\bar{\mathcal{L}}_2(\mathcal{A}) := \mathbb{E}_{\theta \sim \mathcal{P}_2}[\mathcal{L}(\mathcal{A}(\mathcal{D}_\theta); \mathcal{D}_{\theta_1}^{(2)})]$, similar to $\bar{\mathcal{L}}_1(\mathcal{A})$. The overall test loss is given by $\bar{\mathcal{L}}_\mathcal{P}(\mathcal{A}) = r\bar{\mathcal{L}}_1(\mathcal{A}) + (1-r)\bar{\mathcal{L}}_2(\mathcal{A})$. Since $\mathrm{D}^+F_{\mathcal{P}_1}(0) < 0$, we can rearrange (11) to obtain $r\mathrm{D}^+F_{\mathcal{P}_1}(0) - (1-r)\mathrm{D}^-F_{\mathcal{P}_2}(M) > 0$. Intuitively, this means that increasing the capacity assigned to learn $\mathcal{U}_1$ by one unit and reducing the capacity for $\mathcal{U}_2$ by one unit will increase the overall test loss, compared to fully assigning capacity to $\mathcal{U}_2$ and none to $\mathcal{U}_1$. Alternatively, we can view $\frac{r\mathrm{D}^+F_{\mathcal{P}_1}(0)}{(1-r)\mathrm{D}^-F_{\mathcal{P}_2}(M)}$ as the ratio of cost-effectiveness of the knowledge-dense dataset relative to web data. Hence, the model should prioritize web data and not learn from the knowledge-dense dataset when this ratio is below 1.

*Proof.* Let $h$ be the predictor picked by $\mathcal{A}$ on $\mathcal{D}_\theta$. Let $\mathcal{X}_1$ and $\mathcal{X}_2$ be the supports of $x$ in $\mathcal{D}_{\theta_1}^{(1)}$ and $\mathcal{D}_{\theta_2}^{(2)}$, respectively. Let $m_1 := I(h|_{\mathcal{X}_1}; \mathcal{D}_{\theta_1})$ and $m_2 := I(h|_{\mathcal{X}_2}; \mathcal{D}_{\theta_2})$. By data processing inequality, we have

$$m_1 = I(h|_{\mathcal{X}_1}; \mathcal{D}_{\theta_1}) \leq I(h; \mathcal{D}_{\theta_1}),$$
$$m_2 = I(h|_{\mathcal{X}_2}; \mathcal{D}_{\theta_2}) \leq I(h; \mathcal{D}_{\theta_2}),$$

Further noticing that $I(h; \mathcal{D}_\theta) = I(h; \mathcal{D}_{\theta_1}; \mathcal{D}_{\theta_2}) \geq I(h; \mathcal{D}_{\theta_1}) + I(h; \mathcal{D}_{\theta_2})$, we have

$$m_1 + m_2 \leq I(h; \mathcal{D}_\theta) \leq M.$$

Since $h|_{\mathcal{X}_1}$ and $h|_{\mathcal{X}_2}$ are valid predictors on $\mathcal{D}_{\theta_1}$ and $\mathcal{D}_{\theta_2}$, respectively, we have

$$\mathbb{E}[\mathcal{L}(h; \mathcal{D}_{\theta_1})] = \mathbb{E}[\mathcal{L}(h|_{\mathcal{X}_1}; \mathcal{D}_{\theta_1})] \geq F_{\mathcal{P}_1}(m_1),$$
$$\mathbb{E}[\mathcal{L}(h; \mathcal{D}_{\theta_2})] = \mathbb{E}[\mathcal{L}(h|_{\mathcal{X}_2}; \mathcal{D}_{\theta_2})] \geq F_{\mathcal{P}_2}(m_2) \geq F_{\mathcal{P}_2}(M - m_1).$$

Adding the two inequalities with weights $r$ and $1 - r$, we have

$$\bar{\mathcal{L}}_{\mathcal{P}}(\mathcal{A}) = \mathbb{E}[\mathcal{L}(h; \mathcal{D}_\theta)] \geq rF_{\mathcal{P}_1}(m_1) + (1 - r)F_{\mathcal{P}_2}(M - m_1).$$

By convexity (Lemma F.1), we have

$$F_{\mathcal{P}_1}(m_1) \geq F_{\mathcal{P}_1}(0) + \mathrm{D}^+ F_{\mathcal{P}_1}(0)m_1, \qquad F_{\mathcal{P}_2}(M - m_1) \geq F_{\mathcal{P}_2}(M) - \mathrm{D}^- F_{\mathcal{P}_2}(M)m_1.$$

Plugging these into the previous inequality, we have

$$\mathbb{E}[\mathcal{L}(h; \mathcal{D}_\theta)] \geq rF_{\mathcal{P}_1}(0) + (1 - r)F_{\mathcal{P}_2}(M) + \left( r\mathrm{D}^+ F_{\mathcal{P}_1}(0) - (1 - r)\mathrm{D}^- F_{\mathcal{P}_2}(M) \right) m_1.$$

By (11) and the fact that $\mathrm{D}^+ F_{\mathcal{P}_1}(0) \leq 0$, we have $r\mathrm{D}^+ F'_{\mathcal{P}_1}(0) > (1 - r)\mathrm{D}^- F'_{\mathcal{P}_2}(M)$. So the right-hand side is strictly increasing in $m_1$.

Now we claim that $m_1 = 0$. If not, then the following learning algorithm $\mathcal{A}'$ is better than $\mathcal{A}$. Let $\mathcal{A}_1$ be an optimal 0-bounded-capacity learner on $\mathcal{U}_1$ and $\mathcal{A}_2$ be an optimal $M$-bounded-capacity learner on $\mathcal{U}_2$. Run the algorithms to obtain $h_1 \sim \mathcal{A}_1(\mathcal{D}_\theta|_{\mathcal{X}_1})$ and $h_2 \sim \mathcal{A}_2(\mathcal{D}_\theta|_{\mathcal{X}_2})$. Then, whenever seeing an input $x$ from $\mathcal{X}_1$, output $h_1(x)$; otherwise output $h_2(x)$. This algorithm achieves the expected loss $rF_{\mathcal{P}_1}(0) + (1 - r)F_{\mathcal{P}_2}(M)$, which is strictly less than $\bar{\mathcal{L}}_{\mathcal{P}}(\mathcal{A})$ and contradicts the optimality of $\mathcal{A}$.

Therefore, for the optimal algorithm $\mathcal{A}$, $\bar{\mathcal{L}}_1(\mathcal{A}) = F_{\mathcal{P}_1}(0)$. $\qquad\square$

**Lemma F.7.** *Let $\mathcal{U} = (\mathcal{P}, \mathcal{D}_\theta)$ be an orthogonal mixture of $\mathcal{U}_1 = (\mathcal{P}_1, \mathcal{D}_{\theta_1})$ and $\mathcal{U}_2 = (\mathcal{P}_2, \mathcal{D}_{\theta_2})$ with mixing ratio $r$. For all $r \in (0, 1)$, $M \geq 0$ and $\beta \geq 0$, if the following inequality holds,*

$$\frac{r}{1 - r} > \frac{\mathrm{D}^+ F_{\mathcal{P}_2}(M - \beta)}{\mathrm{D}^- F_{\mathcal{P}_1}(\beta)}, \tag{12}$$

*then for any optimal $M$-bounded-capacity learner $\mathcal{A}$ on $\mathcal{U}$, $\mathbb{E}_{\theta \sim \mathcal{P}} \mathbb{E}_{h \sim \mathcal{A}(\mathcal{D}_\theta)}[\mathcal{L}(h; \mathcal{D}_{\theta_1})] \leq F_{\mathcal{P}_1}(\beta)$.*

**Intuitive Explanation.** Similar to the explanation for Lemma F.6, we can rearrange Equation (12) into $-r\mathrm{D}^- F_{\mathcal{P}_1}(\beta) + (1 - r)\mathrm{D}^+ F_{\mathcal{P}_1}(M - \beta) < 0$. Intuitively, this means that reducing the capacity assigned to learn $\mathcal{U}_1$ by one unit and increasing the capacity for $\mathcal{U}_2$ by one unit will increase the overall test loss, compared to assigning capacity $\beta$ to $\mathcal{U}_1$ and $M - \beta$ to $\mathcal{U}_2$. Therefore, the optimal $M$-bounded-capacity learner $\mathcal{A}$ will assign at least capacity $\beta$ to learn $\mathcal{U}_1$, resulting in a test loss on $\mathcal{U}_1$ that is lower than $F_{\mathcal{P}_1}(\beta)$. Alternatively, we can view $\frac{r\mathrm{D}^- F_{\mathcal{P}_1}(\beta)}{(1 - r)\mathrm{D}^+ F_{\mathcal{P}_2}(M - \beta)}$ as the ratio of cost-effectiveness of the knowledge-dense dataset relative to web data. Hence, the model should do its best to learn the knowledge-dense dataset when this ratio is above 1.

*Proof.* Similar to the previous proof, letting $m_1 := I(h|_{\mathcal{X}_1}; \mathcal{D}_{\theta_1})$ and $m_2 := I(h|_{\mathcal{X}_2}; \mathcal{D}_{\theta_2})$, we have

$$\begin{aligned}
m_1 + m_2 &\leq I(h; \mathcal{D}_\theta) \leq M, \\
\mathbb{E}[\mathcal{L}(h; \mathcal{D}_{\theta_1})] &= \mathbb{E}[\mathcal{L}(h|_{\mathcal{X}_1}; \mathcal{D}_{\theta_1})] \geq F_{\mathcal{P}_1}(m_1), \\
\mathbb{E}[\mathcal{L}(h; \mathcal{D}_{\theta_2})] &= \mathbb{E}[\mathcal{L}(h|_{\mathcal{X}_2}; \mathcal{D}_{\theta_2})] \geq F_{\mathcal{P}_2}(m_2) \geq F_{\mathcal{P}_2}(M - m_1), \\
\mathbb{E}[\mathcal{L}(h; \mathcal{D}_\theta)] &\geq rF_{\mathcal{P}_1}(m_1) + (1 - r)F_{\mathcal{P}_2}(M - m_1).
\end{aligned}$$

First, we show that $m_1 \geq \beta$. If not, then by convexity (Lemma F.1), we have

$$\begin{aligned}
F_{\mathcal{P}_1}(m_1) &\geq F_{\mathcal{P}_1}(\beta) - \mathrm{D}^- F_{\mathcal{P}_1}(\beta) \cdot (\beta - m_1), \\
F_{\mathcal{P}_2}(M - m_1) &\geq F_{\mathcal{P}_2}(M - \beta) + \mathrm{D}^+ F_{\mathcal{P}_2}(M - \beta) \cdot (\beta - m_1).
\end{aligned}$$

Plugging these into the previous inequality, we have

$$\mathbb{E}[\mathcal{L}(h; \mathcal{D}_\theta)] \geq rF_{\mathcal{P}_1}(\beta) + (1 - r)F_{\mathcal{P}_2}(M - \beta) + \left( -r\mathrm{D}^- F_{\mathcal{P}_1}(\beta) + (1 - r)\mathrm{D}^+ F_{\mathcal{P}_2}(M - \beta) \right) (\beta - m_1).$$

By (12) and the fact that $\mathrm{D}^- F_{\mathcal{P}_1}(\beta) \leq 0$, we have $r\mathrm{D}^- F_{\mathcal{P}_1}(\beta) < (1 - r)\mathrm{D}^+ F_{\mathcal{P}_2}(M - \beta)$. So the right-hand side is strictly decreasing in $m_1$.

Next, we prove by contradiction that $m_1 \geq \beta$. If $m_1 < \beta$, the following learning algorithm $\mathcal{A}'$ is better than $\mathcal{A}$. Let $\mathcal{A}_1$ be an optimal $\beta$-bounded-capacity learner on $\mathcal{U}_1$ and $\mathcal{A}_2$ be an optimal $(M - \beta)$-bounded-capacity learner on $\mathcal{U}_2$. Run the algorithms to obtain $h_1 \sim \mathcal{A}_1(\mathcal{D}_\theta|_{\mathcal{X}_1})$ and $h_2 \sim \mathcal{A}_2(\mathcal{D}_\theta|_{\mathcal{X}_2})$. Then, whenever seeing an input $x$ from $\mathcal{X}_1$, output $h_1(x)$; otherwise output $h_2(x)$.

This algorithm achieves the expected loss $rF_{\mathcal{P}_1}(\beta) + (1-r)F_{\mathcal{P}_2}(M - \beta)$, which is strictly less than $\bar{\mathcal{L}}_{\mathcal{P}}(\mathcal{A})$.

Therefore, we have $m_1 \geq \beta$ for the algorithm $\mathcal{A}$. Now we prove that $\mathbb{E}[\mathcal{L}(h; \mathcal{D}_{\theta_1})] \leq F_{\mathcal{P}_1}(\beta)$. If not, then the following learning algorithm $\mathcal{A}''$ is better than $\mathcal{A}$. Construct $\mathcal{A}''$ similarly as $\mathcal{A}'$, but with $\mathcal{A}_1$ and $\mathcal{A}_2$ replaced by the optimal $m_1$-bounded-capacity learner on $\mathcal{U}_1$ and the optimal $m_2$-bounded-capacity learner on $\mathcal{U}_2$, respectively. If $\mathbb{E}[\mathcal{L}(h; \mathcal{D}_{\theta_1})] > F_{\mathcal{P}_1}(\beta)$, then $\mathcal{A}''$ achieves a lower expected loss than $\mathcal{A}$, which contradicts the optimality of $\mathcal{A}$. $\qquad\square$

Now we consider the case where $\mathcal{U}_1$ is a factual data universe, and $\mathcal{U}_2$ is an arbitrary data universe.

**Theorem F.8.** *Let $\mathcal{U}_1$ be a factual data universe with $K$ random facts, each with the same exposure frequency $p$, and the entropies of their target tokens sum to $H_{\text{tot}} := \sum_{i=1}^{K} H(\mathcal{Y}_i)$. Let $\mathcal{U}_2$ be an arbitrary data universe. Let $\mathcal{U} = (\mathcal{P}, \mathcal{D}_\theta)$ be an orthogonal mixture of $\mathcal{U}_1$ and $\mathcal{U}_2$ with mixing ratio $r$. For all $r \in (0, 1)$ and $M \geq 0$,*

1. *if $\frac{r}{1-r} \cdot p < -\mathrm{D}^- F_{\mathcal{P}_2}(M)$, then $\bar{\mathcal{L}}_1(\mathcal{A}) = F_{\mathcal{P}_1}(0)$;*

2. *if $\frac{r}{1-r} \cdot p > -\mathrm{D}^+ F_{\mathcal{P}_2}(M - H_{\text{tot}})$, then $\bar{\mathcal{L}}_1(\mathcal{A}) = F_{\mathcal{P}_1}(\infty)$.*

*Proof.* By Theorem F.3, $\mathrm{D}^+ F_{\mathcal{P}_1}(0) = \mathrm{D}^- F_{\mathcal{P}_1}(H_{\text{tot}}) = p$. Plugging this into Lemma F.6 and Lemma F.7 with $\beta = H_{\text{tot}}$ finishes the proof. $\qquad\square$

Now we are ready to prove the main theorem we stated in Section 4.4. Recall that

$$M_0^-(t) := \sup\{M \geq 0 : -F'_{\mathcal{P}_2}(M) > t\},$$
$$M_0^+(t) := \inf\{M \geq 0 : -F'_{\mathcal{P}_2}(M) < t\},$$

**Theorem F.9** (Theorem 4.3, restated)**.** *For any optimal $M$-bounded-capacity learner $\mathcal{A}$,*

1. *if $M \leq M_0^-(\frac{r}{1-r} \cdot p)$, then $\bar{\mathcal{L}}_1(\mathcal{A}) = F_{\mathcal{P}_1}(0)$;*

2. *if $M \geq M_0^+(\frac{r}{1-r} \cdot p) + H_{\text{tot}}$, then $\bar{\mathcal{L}}_1(\mathcal{A}) = F_{\mathcal{P}_1}(\infty)$.*

*Proof.* This is a direct consequence of Theorem F.8 by noting that (1) $-\mathrm{D}^- F_{\mathcal{P}_2}(M)$ is left continuous and non-increasing in $M$; (2) $-\mathrm{D}^+ F_{\mathcal{P}_2}(M)$ is right continuous and non-increasing in $M$; (3) $F_{\mathcal{P}_2}(M)$ is almost everywhere differentiable. $\qquad\square$

