# OpenReview forum: "Data Mixing Can Induce Phase Transitions in Knowledge Acquisition"
_NeurIPS.cc/2025/Conference — NeurIPS 2025 spotlight_

### Official Review · Reviewer_DKRZ · 2025-07-02

**Clarity:** 3
**Significance:** 2
**Originality:** 2
**Rating:** 4
**Confidence:** 4

**Summary:**

In this paper, authors investigate how mixing small amounts of high-quality data such as synthetic biographies with web-scraping data affects knowledge acquisition. Through controlled experiments. authors discover two types of phase transitions: along model size an along mixing ratio. The authors then attribute these sharp transitions to a capacity allocation phenomenon. They also formalize their intuition to derive a power-law relationship that accurately predicts the critical thresholds.

**Questions:**

1. Authors claim they observe "phase" transitions across model scale and data ratio. But what qualifies as a phase transition versus a linear development? Without a frame of reference, how do we know the observe performance change is sharp and considers as a phase transition?

**Ethical Concerns:**

["NO or VERY MINOR ethics concerns only"]

**Final Justification:**

Reviewers have thoroughly addressed my concerns during rebuttal and I think overall the study provide a comprehensive analysis on scaling and data mixing.

**Limitations:**

No, authors adequately addressed the limitations and potential negative societal impact of their work.

**Paper Formatting Concerns:**

Many pages—particularly those starting with plots—appear to have significantly smaller top margins than expected. Additionally, the spacing between sections feels overly tight throughout. This gives the strong impression that the authors may have manually adjusted the margins or spacing to fit within the page limit.

**Quality:**

3

**Strengths And Weaknesses:**

**Strengths**

1. Data mixing during pretraining is a critical and under-explored topic, and this paper addresses it using a highly controlled and well-designed experimental setup. By systematically varying the model size and the ratio of knowledge-dense to web data, the authors reveal two sharp phase transitions in knowledge acquisition. The clarity of these findings, and the effective presentation of results (e.g., Figures 1–3), make the empirical contribution compelling.


2. In addition to analysis, authors also propose two actionable and well-motivated strategies to improve knowledge retention task. Overall, this paper has a complete analysis of theory, empirical findings and practical insights.


**Weakness**

1. Prior work (e.g., [1], [2]) has shown that phase transitions and emergent behaviors in LLMs can be artifacts of (a) metric thresholding and (b) sensitivity to random initialization. However, this paper does not adequately rule out those possibilities. For instance, all key evaluations use discrete accuracy measures (e.g., exact match of factual answers). It would strengthen the case considerably if the authors provided results using more continuous metrics (e.g., token-level F1, log-likelihood) and ran models with multiple random seeds to test robustness against training noise.


2. Figures 1, 2, and 5 use a mix of linear and log scales on the y-axis—sometimes for the same metric (accuracy) across different plots. This inconsistency is not explained and raises the concern that log scaling may be exaggerating the sharpness of transitions. A clearer and more principled explanation of axis choices would help ensure the visualizations do not unintentionally bias interpretation.



3. One of the motivations for data mixing is to enable multitask performance (e.g., reasoning, coding, factual recall). However, in this paper, the “web” data serves more as a placeholder than a concrete "competing" task (i.e., learning synthetic biography versus math). As a result, the paper does not deeply analyze trade-offs between capabilities caused by data mixes—for example, whether increasing knowledge-dense data harms performance on reasoning.


4. Although Section 6 introduces two mitigation strategies, both seem most applicable to factoid-style datasets like synthetic biographies, where each data point is short, modular, and isolated. It is less clear how these strategies would extend to richer forms of knowledge or reasoning, where facts are embedded in longer contexts or interconnected. This limits the generality of the proposed solutions.



[1] Are Emergent Abilities of Large Language Models a Mirage?   http://arxiv.org/abs/2304.15004


[2] Distributional scaling for emergent capabilities. https://arxiv.org/abs/2502.17356

---

> ### Author Rebuttal · Authors · 2025-07-31
>
> We sincerely thank the reviewer for the thoughtful feedback and for acknowledging that our paper presents a well-controlled study on an important and under-explored topic. We are glad that the reviewer found our empirical contributions compelling and our proposed strategies to be well-motivated and actionable. Below, we address the reviewer's concerns and questions.
>
> ---
> **W1:** Prior work has shown that phase transitions and emergent behaviors in LLMs can be artifacts of ... It would strengthen the case considerably if the authors provided results using more continuous metrics (e.g., token-level F1, log-likelihood) and ran models with multiple random seeds to test robustness against training noise.
>
> **A:** 1. **Phase transitions in validation loss**: In addition to accuracy, **we do observe phase transitions in SynBio validation loss**, a continuous metric. As shown in Table 1 (a) below (corresponding to $r=0.1$ in Figure 1), the loss remains nearly flat as the model size increases from 14M to 160M, decreasing by only 0.03. However, a sharp drop of 0.45 occurs between 256M and 410M, aligning with the sudden jump in accuracy from 0.9% to 80.3%. Similar phase transitions in validation loss can be observed in Table 1(b) (from Figure 2(b)). These trends confirm that **the observed phase transition is not merely an artifact of accuracy thresholding, but also reflected in smooth metrics like loss**.
>
> **Table 1 (a) Validation loss and accuracy on SynBio-320k for training runs in Figure 1 with $r=0.1$. Note: 0.4% accuracy $\approx$ random guessing.**
>
> | Model size (M)| Val. loss on SynBio-320k | Acc (%)|
> |:---:|:---:|:---:|
> |14 | 0.86    |0.4      |
> |31  | 0.86  | 0.4     |
> |70 | 0.85   | 0.4     |
> |120 | 0.85 | 0.4     |
> |160  | 0.83   | 0.4     |
> |256  | 0.78 | 0.9     |
> |410   | 0.33   | 80.3     |
>
> **Table 1 (b) Validation loss and accuracy on SynBio-1.28M for training runs in Figure 2 (b). Note: 0.4% accuracy $\approx$ random guessing.**
> | Mixing ratio $r$ | Val. loss on SynBio-1.28M| Acc (%)|
> |:------------:|:--:|:-------:|
> |0 | 0.86 | 0.4     |
> |0.1 | 0.86 | 0.4     |
> |0.2  | 0.85  | 0.4     |
> |0.3 | 0.85 | 0.4     |
> |0.4   | 0.33| 80.4    |
>
>
> 2. **Robustness to training noise**. We replicate the experiments in Figure 2 (a) with two additional random seeds and report the mean and standard deviation below. While accuracy varies slightly with random seeds, the phase transition behavior remains consistent and clearly observable across runs.
>
> **Table 2: Mean and standard deviations over 3 random seeds for runs in Figure 2 (a).**
>
> | Mixing ratio $r$| Mean acc (%)|Std. dev. (%)|
> |:------------:|:-------:|:-------:|
> |0.1           |0.4      |0.0|
> |0.15          | 0.4     |0.0|
> |0.2           |0.4      |0.0|
> |0.25          |0.8      |0.0|
> |0.3           |7.3      |2.8|
> |0.35          | 27.5     |2.8|
> |0.4           |40.4    |3.1|
> |0.45          |58.1    |3.1|
>
> We will include validation loss results and report experiments with multiple random seeds in the next version to strengthen our findings.
>
> ---
> **W2:** Figs. 1, 2, 5 use a mix of linear and log scales on the y-axis—sometimes for the same metric (acc.) across different plots... A clearer and more principled explanation of axis choices would help ensure the visualizations do not unintentionally bias interpretation.
>
> **A:** Our principle for axis choices in Figures 1, 2, and 5 is to **match the scaling of the x- and y-axes, in order to avoid distorting trends**.
>
> - For phase transitions with respect to mixing ratio $r$, we use a linear x-axis, since $r$ is a tunable parameter with uniform meaning across its range. Accordingly, we keep the y-axis linear.
>
> - For phase transitions with respect to model size, we use a logarithmic x-axis, as model size typically increases exponentially in practice (e.g., via width or depth). To avoid misleading exponential trends due to this scaling, we also apply a log scale on the y-axis, so linear accuracy growth does not appear exaggerated. This ensures that the sharp transition we observe is not an artifact of axis distortion.
>
> As additional support, we provide the exact data points for $r = 0.2$ from Figure 1 below. The reviewer is welcome to replot the curve and will find that the sharp transition remains clearly visible even when changing the y-axis from log scale to linear. If plotting with a log scale, we recommend using a symmetric log scale with a linear threshold of 1, to avoid exaggeration near zero.
>
> **Table 3: Data points for $r=0.2$ in Figure 1**
>
> |Model size (M) |Acc. (%)|
> |:-------------:|:------:|
> |   14   |    0.4    |
> |   31 |     0.4   |
> |   70 |     0.4   |
> |    95  |      0.5  |
> |     120  |      5.2  |
> |     160 |      82.1  |
> |     256|       93.1 |
> |     410   |       97.6 |
>
> We will include a clear explanation of our axis scaling choices in our next version to avoid potential confusion.
>
> ---
>
> **W3**: One of the motivations for data mixing is to enable multitask performance (e.g., reasoning, coding, factual recall) ... As a result, the paper does not deeply analyze trade-offs between capabilities caused by data mixes, for example, whether increasing knowledge-dense data harms performance on reasoning.
>
> **A:**  We agree that multitask learning is a key motivation for data mixing, and that different capabilities may compete for model capacity. In fact, **Sec. 6 of our paper explicitly discusses this trade-off.**
>
> - As stated at the beginning of Sec. 6, our motivation for proposing the two strategies is to mitigate this dilemma: a small $r$ limits learning from knowledge-dense data, while a large $r$ may degrade the general capabilities acquired from web-scale data (e.g., commonsense reasoning, text understanding). Our strategies aim to enhance knowledge acquisition even at small $r$, thereby preserving overall model capability.
>
> - All experiments in Section 6 not only report the accuracy on knowledge-dense data, but also the validation loss on web data and downstream performance (averaged over five benchmarks) to assess general capabilities. For example, Figure 7 (b) and Tables 3 and 4 show that
>     -  Without our strategies, increasing $r$ worsens validation loss on Pile and degrades downstream performance.
>
>     - Applying either strategy enables the model to acquire significantly more factual knowledge at a small mixing ratio ($r = 0.1$) while maintaining Pile loss and downstream task performance.
>
> In our next version, we will better highlight this tradeoff between knowledge acquisition and general capabilities and clarify how our strategies help mitigate it.
>
> ---
> **W4:** It is less clear how these strategies (in Sec. 6) would extend to richer forms of knowledge or reasoning, where facts are embedded in longer contexts or interconnected. This limits the generality of the proposed solutions.
>
> **A:** We thank the reviewer for pointing out the important direction to test our mitigation strategies on richer forms of knowledge or reasoning. However, we would like to clarify the intended scope and generality of our contributions:
>
> 1. Our main contributions are to discover and formulate the phase transition in knowledge acquisition due to data mixing, along with theoretical analyses.
>
> 2. The two mitigation strategies in Section 6 are practical implications of our theory: by increasing the marginal value (i.e., cost-effectiveness) of the knowledge we care about, we can significantly improve learning outcomes. While these strategies are demonstrated on factoid-style data, the underlying principle is broadly applicable.
>
> 3. For example, random subsampling could be adapted to other domains by selectively reducing the volume of less relevant data. CKM) could be extended by prompting LLMs to rephrase or compress richer knowledge into more concise formats.
>
> Therefore, we leave it as future work to explore the extensions of our mitigation strategies to more complex tasks.
>
> ---
>
> **Q1:** What qualifies as a phase transition versus a linear development? Without a frame of reference, how do we know the observed performance change is sharp and considered a phase transition?
>
> **A:** Thank you for this thoughtful question. Let $x$ be the independent variable (e.g., model size or mixing ratio), and $y \in [0, 1]$ be the dependent variable (e.g., accuracy).
>
> - **Phase transition:** As defined in our abstract and introduction, we consider $y$ to exhibit a phase transition with respect to $x$ if there exists a threshold $x_\mathrm{thres}$ and a small constant $\delta$ such that for $x < x_\mathrm{thres}$, $y$ remains below $\delta$ with nearly no improvement, but for $x > x_\mathrm{thres}$, $y$ rises rapidly.
>
> - **Linear development:** In contrast, if $y$ increases gradually and proportionally with $x$ before saturating at 1, we consider it a linear trend. The curve would appear as a steady incline without abrupt jumps.
>
> To illustrate this contrast, below is an example of what a linear counterpart to Figure 2(b) might look like:
>
> **Table 4: A hypothetical linear counterpart to Figure 2(b).**
>
> | Mixing ratio $r$ | Acc (%)|
> |:-----:|:-----:|
> |0   | 0.4    |
> |0.1 | 19.5   |
> |0.2   | 42.3   |
> |0.3 | 57.9    |
> |0.4 | 80.4    |
>
> We will revise the paper to explicitly clarify the definition of phase transitions, contrast them with linear trends, and explain how our empirical results (e.g., in Figure 2(b)) fit this definition.
>
> ---
> **Response to format concerns:** We are sorry about that. We will revise the format and move many of the details in Section 6 to the appendix.
>
> ---
> We thank the reviewer again for the thoughtful and constructive feedback! We hope our responses have addressed all questions from the reviewer. If so, we kindly ask the reviewer to raise the rating. If the reviewer has any further questions, we will be more than happy to answer them.

---

### Official Review · Reviewer_rq2w · 2025-07-02

**Clarity:** 3
**Significance:** 3
**Originality:** 3
**Rating:** 5
**Confidence:** 3

**Summary:**

This paper investigates how mixing knowledge-dense data with large-scale web data during LLM pre-training can cause *phase transitions* in knowledge acquisition. Using synthetic and real-world datasets, the authors show that knowledge retention does not always scale smoothly with model size or data mix ratio: instead, there are critical thresholds below which models memorize almost nothing, and above which they suddenly acquire much more knowledge. They develop an information-theoretic framework to explain this as a capacity allocation problem, and they validate it with empirical results and scaling laws. The paper also proposes practical strategies—like random subsampling and compact rephrasing—to improve knowledge retention under low mixing ratios.

**Questions:**

1. Based on Weakness 2, could you design a task that better aligns with the definition of reasoning to more effectively support your claims?
2. Although your experimental setup follows prior work, I think this setting—consisting of only a single entity and its associated attributes—is insufficient to verify whether your conclusions hold in scenarios involving multiple entities and more complex relationships. Could you consider supplementing your study with some simple experiments under a more complex setting?
3. The experiments in Section 6 appear to be conducted only on a 410M parameter model, which is not commonly used in current real-world scenarios. Could you reproduce this part of the study on a model with approximately 7B parameters to better demonstrate the generalizability of your applications?

**Ethical Concerns:**

["NO or VERY MINOR ethics concerns only"]

**Final Justification:**

This paper presents extensive theoretical and empirical work revealing phase transitions in LLMs’ knowledge acquisition when mixing data types, offering practical insights and strategies for effective pre-training at scale. During rebuttals, the authors have addressed my main concerns by providing results on more complex tasks and additional models. Therefore, I have raised my score from 4 to 5.

**Limitations:**

Yes

**Quality:**

4

**Strengths And Weaknesses:**

### Strengths

1. The paper demonstrates a substantial amount of work, including comprehensive experiments, thorough theoretical analysis, and corresponding application designs.
2. The work presents many interesting findings, which have clear implications for data curation and training strategies for large language models at different scales.

### Weaknesses

1. The spacing between section titles and the main text is too narrow, which does not conform to the standard template. As a result, the overall content appears overly dense, negatively affecting readability.
2. I believe the design of the Reasoning Tasks in Section 3.3 is questionable. Although the task appears to assess a slope-related problem involving mathematical reasoning, the evaluation instances frequently appear in the training data. As a result, the model can rely purely on memorization to answer the slope questions correctly, rather than genuinely computing the slope through intermediate reasoning steps. Therefore, in my opinion, this task design does not align with the fundamental definition of reasoning and remains within the scope of factual memorization.

---

> ### Author Rebuttal · Authors · 2025-07-31
>
> We sincerely thank the reviewer for the positive feedback and for acknowledging the thoroughness of our work, the interesting findings, and the clear implications for data curation and LLM training strategies. Below, we address the reviewer's questions and concerns.
>
>
> ---
>
> **W1:** The spacing between section titles and the main text is too narrow.
>
> **A:** Sorry about that. We will revise the formatting to improve readability and move many of the details in Section 6 to the appendix.
>
> ---
>
> **W2+Q1:** I believe the design of the Reasoning Tasks in Section 3.3 is questionable. The model can rely purely on memorization to answer the slope questions correctly, rather than genuinely computing the slope through intermediate reasoning steps.
>
> **A:** **We respectfully disagree with the concern that the model could rely purely on memorization to solve the slope questions.** The total number of possible slope calculation problems in our setup is $100 \times 100 \times 99 \times 100 = 9.9 \times 10^7$. In contrast, a typical training run in Figure 5(b) with $r = 0.4$ sees fewer than $3.5 \times 10^6$ unique slope examples, less than 5% of the full space. Despite this limited coverage, the model still achieves 60% accuracy at test time, where examples are uniformly sampled from the full distribution.
>
> This substantial generalization beyond the training data suggests that **the model is indeed learning a generalizable computation procedure, rather than memorizing specific input-output pairs.** We will clarify this point in the next version by explicitly reporting the size of the problem space and the number of unique training samples to avoid potential confusion.
>
> ---
>
> **Q1:** Could you design a task that better aligns with the definition of reasoning to support your claims more effectively?
>
> **A:** Certainly! We design a new task named "max over n numbers". Compared to the slope calculation task, this task has a vastly larger input space, making memorization infeasible.
>
> Specifically, the model is asked to find the maximum number given a list of $n=30$ numbers, each randomly sampled from $\{0, 1, ..., 99\}$. The training samples are formatted as follows:
>
> > Q: Find the maximum value of the following list:
> >
> > [47, 83, 38, ...]
> >
> > Think step-by-step.
> >
> > A:
> >
> > Compare 47 and 83. Keep 83.
> >
> > Compare 83 and 38. Keep 83.
> >
> > ...
> >
> > Final answer: 83.
>
> Following the setup in Section 3.3, we add 3M tokens of such examples to the OpenWebMath dataset (accounting for less than 0.02% of the original OpenWebMath token count) to create a modified version. We then train Pythia models on a mixture of FineWeb-Edu and the modified OpenWebMath dataset, with $r$ denoting the mixing ratio of the modified OpenWebMath.
>
> For evaluation, we generate 1,000 test samples of "max over 30 numbers" and assess whether the model outputs the correct final answer.
>
> As shown in Table 1:
>
> 1. The 70M model achieves a test accuracy of 60% with fewer than 3,500 unique training examples (at $r = 0.5$), a negligible fraction of the total sample space, which has size $10^{60}$. This clearly demonstrates generalization beyond memorization.
>
> 2. We observe similar phase transitions with respect to both model size and mixing ratio, consistent with our main findings.
>
> **Table 1: We observe similar phase transitions in mixing ratio $r$ and model size for the "max over 30 numbers" task.**
>
> **Table 1 (a)：Phase transition in $r$ on 70M Pythia models.**
>
> |Mixing ratio $r$ | Acc. on "max over 30 numbers" (%)|
> |:-----------------: | :----------------------:  |
> |0.1                | 0.0                       |
> |0.2                | 0.0                      |
> |0.3                | 0.5                       |
> |0.4                | 34.6                       |
> |0.5                | 61.8                       |
>
>
>
> **Table 1 (b)：Phase transition in model size at $r$=0.2.**
>
> |Model size (M) | Acc. on "max over 30 numbers" (%)|
> |:-----------------:  | :----------------------:  |
> |31                | 0.0                       |
> |50                | 0.0                      |
> |70                | 0.0                       |
> |95               | 0.1                     |
> |120                | 0.1                       |
> |160                | 2.6                       |
> |256                | 30.5                      |
> |410               | 87.0                    |
>
>
>
>
>
>
>
> ---
>
> **Q2:** The experiment setting is insufficient to verify whether your conclusions hold in scenarios involving multiple entities and more complex relationships. Could you consider supplementing your study with some simple experiments under a more complex setting?
>
>
> **A:** We believe the tasks used in our experiments—slope calculation and "max over n numbers"—already involve multiple entities and require learning complex relationships among them. Specifically, each number can be viewed as an individual entity. The "max over n numbers" task requires the model to learn pairwise comparisons and reason over a sequence of such relations to identify the maximum. The slope calculation task involves both subtraction and division, which represent structured relationships between three numerical entities. For example, subtraction captures the relationship among the minuend, subtrahend, and difference. These tasks go beyond simple recall and reflect nontrivial multi-step reasoning.
>
>
>
>
> ---
>
> **Q3:** The experiments in Section 6 appear to be conducted only on a 410M parameter model, which is not commonly used in current real-world scenarios. Could you reproduce this part of the study on a model with approximately 7B parameters?
>
>
> **A**: Sure! We conduct additional experiments on Pythia 6.9B to show the effectiveness of random subsampling on larger models. Specifically, we follow the continual pre-training setup in Figure 8(b) and apply random subsampling at mixing ratio $r=0.2$. Let  $\rho$ denote the subsampling ratio. As shown in Table 2, setting $\rho$ as 0.5 and 0.25 significantly improves the model's accuracy on SynBio from 0.4% to 19.5% and 17.5% respectively. This demonstrates that random subsampling remains highly effective even at the 7B scale.
>
> **Table 2: Random subsampling enables the 6.9B model to learn significantly more from knowledge-dense data.**
>
> | Subsampling ratio $\rho$ | Acc. on SynBio-10k (%) |
> |:----------------------:|:---------------:|
> | 1 (no subsampling)                | 0.4           |
> | 0.5                | 19.5          |
> | 0.25                 | 17.9          |
>
>
> ---
>
> We thank the reviewer again for the thoughtful and constructive feedback! We hope our responses have fully addressed the reviewer's concerns. If so, we kindly ask the reviewer to raise the rating. If the reviewer has any further questions, we will be more than happy to answer them.

---

> > ### Author Response · Authors · 2025-08-08
> >
> > Thank you again for taking the time to review our paper and for your valuable feedback!
> >
> > We believe we have addressed all of your questions and concerns in the rebuttal, specifically regarding (1) the design of the slope calculation task, (2) additional experiments on other reasoning tasks, (3) the applicability of our conclusions to more complex scenarios, and (4) experiments with 7B models to evaluate the effectiveness of our mitigation strategies. For points (2) and (4), we have conducted new experiments and included the results in the rebuttal.
> >
> > As the author-reviewer discussion phase is nearing its deadline, we wanted to kindly check whether you have any remaining questions or concerns. We’d be happy to further clarify or elaborate on any aspect of our work.

---

### Official Review · Reviewer_gxsV · 2025-07-02

**Clarity:** 3
**Significance:** 3
**Originality:** 3
**Rating:** 5
**Confidence:** 4

**Summary:**

This paper studies the dynamics of knowledge acquisition in large language model. Specifically, this work examines how things change with different mixtures of data. It aims to address what is learned from small amounts of knowledge rich data and large amounts of less rich data. The authors propose a scaling law that predicts phase transitions in the learning along the axis of data scaling.

**Questions:**

My questions are listed with my weaknesses above.

I'm excited to engage with the authors to clear up the aspects I don't fully understand and I'm optimistic that with some iteration this paper can be made stronger.

**Ethical Concerns:**

["NO or VERY MINOR ethics concerns only"]

**Final Justification:**

The rebuttal was thorough and convincing. This is a good paper and I recommend it be accepted.

**Limitations:**

I'd like to see a bit more discussion on the limitations of experiments. The very controlled setting here allows for nice experimentation, but the authors should better describe and discuss how these findings may or may not translate to real settings where data isn't necessarily broken up so nicely into general data and specific data. Can the authors further explain which of their findings might relate to real world training regimes?

**Paper Formatting Concerns:**

Top margins are too small.

**Quality:**

3

**Strengths And Weaknesses:**

I like the framing and the angle of this paper. I think it addresses a timely and important question. I do have some concerns about clarity and experimental design.

Strengths:
- This work touches on an important topic and makes a reasonable contribution toward trying to deepen our understanding of knowledge acquisition in large language models.
- To my knowledge, this work is novel and original. The controlled experiments on data mixture and the isolation of knowledge-dense data from more general data is interesting.
- The writing is generally good, although I do have some clarity questions below.

Weaknesses:
- Clarity
  - The words "memorized" and "learned" are used in confusing ways in this paper. Can the authors clarify how these things are measured and whether they mean the same thing or not?
  - In the summary of Finding 1 on the second page, the authors write "... the model suddenly memorizes most biographies. Can the authors give a little more description of what "suddenly" means and what an alternative would look like?
- Formatting
  - The margins over the figures on pages 2, 4, 5, 7, and 8 are smaller than the formatting rule allow. Please fix this.
- Experimental Design
  - In Section 2, the training setup describes multiple epochs of some data and only one epoch of the web data. Can the authors address how repeating data may be at play here? Is that a confounder? Is there a possibility that repeating the general data or not repeating the biography data would change the results?
  - Synthetic data poses a potential problem here. This introduces some distillation potential if the synthetic data is generated with an LLM.
  - The Compact Knowledge Mixing is not so clear to me. Can the authors make a bit more clear how the frequencies are changed? Is the data compressed to make it fewer tokens and thus more frequent in the training set? Adding data but keeping the ratio fixed is confusing to me.

---

> ### Author Rebuttal · Authors · 2025-07-31
>
> We sincerely thank the reviewer for the positive and encouraging feedback. We are glad that the reviewer appreciates the significance and originality of this paper. Below, we address the reviewer's questions and concerns.
>
> ---
> **Clarity Concern 1:** The words "memorized" and "learned" are used in confusing ways in this paper. Can the authors clarify how these things are measured and whether they mean the same thing or not?
>
> **A:** This paper mainly studies factual knowledge acquisition, where the model is tasked with recalling the facts seen during training. We use the words "memorized" and "learned "interchangeably to refer to this factual recall ability.  We will clarify this terminology in the next version to avoid confusion.
>
> ---
>
> **Clarity Concern 2:** In the summary of Finding 1 on the second page, the authors write "... the model suddenly memorizes most biographies. Can the authors give a little more description of what "suddenly" means and what an alternative would look like?
>
> **A:** Thank you for this thoughtful question. Below, we provide a clearer contrast between the phase transition behavior we describe and a more gradual, linear trend.
>
> Let $x$ be the independent variable (e.g., model size or mixing ratio), and $y \in [0, 1]$ be the dependent variable (e.g., accuracy).
>
> * **Phase transition:** We say that $y$ exhibits a phase transition with respect to $x$ if there exists a threshold $x_\mathrm{thres}$ and a small constant $\delta$ such that for $x < x_\mathrm{thres}$, $y$ remains close to zero (i.e., below $\delta$) with nearly no improvement. However, once $x$ exceeds the threshold $x_\mathrm{thres}$, $y$ increases sharply within a narrow range of $x$. **This sharp, non-linear change is what we refer to as "sudden"—the model transitions from recalling almost nothing to recalling most facts in just a small increase of $x$**.
>
> * **Alternative: a linear trend:** In contrast, if $y$ increases gradually and roughly proportionally with $x$ before saturating at 1, we consider that a linear or smooth development. The curve would resemble a steady incline, without abrupt jumps in performance.
>
>
> To illustrate this contrast, we present the exact data points in Fiugure 2 (b) and a hypothetical linear counterpart below:
>
>
> **Table 1 (a): Acc. on SynBio-1.28M for training runs in Figure 2 (b). Note: 0.4% accuracy $\approx$ random guessing.**
> | Mixing ratio $r$ | Acc. (%)|
> |:------------:|:-------:|
> |0             |  0.4     |
> |0.1             | 0.4     |
> |0.2             |  0.4     |
> |0.3             |0.4     |
> |0.4             | 80.4    |
>
>
>
>
> **Table 1 (b): A hypothetical linear counterpart to Figure 2(b).**
>
> | Mixing ratio $r$ | Acc. (%)|
> |:----------:|:-------:|
> |0                | 0.4    |
> |0.1             | 19.5   |
> |0.2             | 42.3   |
> |0.3             | 57.9    |
> |0.4             | 80.4    |
>
> The steady progression in Table 1 (b) differs significantly from the actual results in Figure 2(b), where accuracy stays near zero for $r\leq 0.3$ and then jumps abruptly to 80.4% for $r=0.4$.
>
> We will revise our wording in the paper to make the use of "suddenly" more precise in the next version.
>
> ---
>
> **Formatting Concern:** The margins over the figures on pages 2, 4, 5, 7, and 8 are smaller than the formatting rule allow. Please fix this.
>
> **A:** Sorry about that. We will revise the formatting and move many of the details of Section 6 to the Appendix.
>
>
> ---
>
> **Experimental Design Concern 1:** In Section 2, the training setup describes multiple epochs of some data and only one epoch of the web data. Can the authors address how repeating data may be at play here? Is that a confounder? Is there a possibility that repeating the general data or not repeating the biography data would change the results?
>
> **A:**  The choice to train on SynBio for multiple epochs while using only one epoch of web data is intended to **better reflect practical training practices**:
>
> - Web data is abundant, typically comprising trillions of tokens, so a single pass is generally sufficient.
>
> - Knowledge-dense data is much smaller. For instance, Wikipedia contains only a few billion tokens. Mdels often struggle to absorb all the knowledge in a single pass [1, 2]. As a result, it is now common to rephrase such data in to diverse forms to expose the model to the same knowledge multiple times [2, 3].
>
> - Similar to the rephrasing practice, the model does not see exact repetitions of SynBio biographies. Instead, each time a biography is sampled, its five sentences are randomly shuffled, and new sentence templates are sampled on-the-fly for each attribute (as described in Section 2). If we do not expose the model to the biographies multiple times, it fails to memorize any of them.
>
>
>
>
>
>
> [1] Jing Huang, Diyi Yang, and Christopher Potts. Demystifying verbatim memorization in large language models.
>
> [2] Kimi Team. Kimi K2: Open Agentic Intelligence.
>
> [3] Xintong Hao, Ke Shen, and Chenggang Li. MAGA: MAssive Genre-Audience Reformulation to Pretraining Corpus Expansion.
>
> ---
>
> **Experimental Concern 2:** Synthetic data poses a potential problem here. This introduces some distillation potential if the synthetic data is generated with an LLM.
>
> **A:** As described in Section 2, our synthetic biographies are generated by (1) randomly sampling a value for each attribute of a person, and (2) converting each (name, attribute, value) triple into natural text using sentence templates listed in Table 6. While these templates were created via prompting GPT-4o and then manually polished, they are **simple surface-level rephrasings** and **do not introduce any additional knowledge from GPT-4o**. Therefore, **no distillation occurs**. The core challenge lies in learning the correct mapping from (name, attribute) to value, not in modeling or memorizing the template text.
>
>
> ---
>
> **Experimental Concern 3:** The Compact Knowledge Mixing is not so clear to me. Can the authors make a bit more clear how the frequencies are changed? Is the data compressed to make it fewer tokens and thus more frequent in the training set? Adding data but keeping the ratio fixed is confusing to me.
>
>
>
>
> **A:** Yes, CKM increases the frequency of each fact by reducing its representation length. In Figure 7(b), $\tau = 0.1$, $0.3$, and $0.6$ correspond to roughly 2×, 3×, and 4× increases in fact frequency, respectively.
>
> The “adding data but keeping the ratio fixed” setup is implemented as follows: A mixing ratio \$r\$ means that \$r\$ fraction of training tokens comes from WikiBio. When we apply CKM, we augment WikiBio by adding compact tuple representations. To maintain the same token budget for WikiBio after augmentation, we reduce the number of epochs over it proportionally. Although the model sees fewer total passes over the dataset, each individual fact appears more frequently per epoch, thereby increasing its exposure.
>
> We will clarify how CKM changes fact frequency and explicitly describe this implementation detail in the next version.
>
>
> ---
>
> **More discussions on limitations:** I'd like to see a bit more discussion on the limitations of experiments. The very controlled setting here allows for nice experimentation, but the authors should better describe and discuss how these findings may or may not translate to real settings where data isn't necessarily broken up so nicely into general data and specific data. Can the authors further explain which of their findings might relate to real-world training regimes?
>
> **A:** We agree that real-world data is not always cleanly separable into “general” and “specific” categories. However, our findings still apply at the level of individual knowledge pieces. That is, a specific fact or reasoning procedure may not be acquired at all if its frequency or the model size falls below a threshold. This core insight generalizes to real-world training scenarios, which is supported by the following evidence:
>
> - Theoretical evidence: Section 4 derives a threshold frequency for the acquisition of each individual fact.
>
> - Empirical evidence 1: Figures 6(c) and 10 show that our predicted power-law threshold holds for real-world Wikipedia facts.
>
> - Empirical evidence 2: Section 3.3 and additional experiments (see response to Reviewer rq2w's Q1) demonstrate that reasoning procedures—such as slope calculation and max-over-n—can also exhibit sharp transitions similar to factual knowledge.
>
> We appreciate the reviewer’s insightful question and will incorporate this discussion into the next version to better clarify how our findings extend to practical training regimes.
>
> ---
>
> We thank the reviewer again for the thoughtful and constructive feedback! We hope our responses have addressed all questions from the reviewer. If so, we kindly ask the reviewer to raise the rating. If the reviewer has any further questions, we will be more than happy to answer them.

---

> > ### Comment · Reviewer_gxsV · 2025-08-03
> > **Nice replies**
> >
> > The authors thoroughly responded to my review and clarified all the points I had asked about. I'm increasing my score accordingly.
> >
> > Thank you!

---

> > > ### Author Response · Authors · 2025-08-03
> > > **Thank you!**
> > >
> > > Thank you so much for the inspiring reply! Your comments demonstrated deep engagement and great care, which we are truly grateful for.
> > >
> > > Your question about the connection to real-world settings prompted us to reflect more deeply on the practical implications of our findings. Your comments on the definition of phase transitions, on the multi-epoch setting, and on the implementation of CKM helped us clarify key concepts and improve the writing, making our intentions clearer to readers.
> > >
> > > Thank you again for your support and for helping us strengthen the paper!

---

### Official Review · Reviewer_N1QD · 2025-07-03

**Clarity:** 2
**Significance:** 3
**Originality:** 3
**Rating:** 4
**Confidence:** 2

**Summary:**

This paper studies the relationship between model training and dataset mixing, model size, revealing some interesting phenomenon, along with some theoretical analysis.

**Questions:**

The paper has shown that the knowledge acquisition happens only when mixing ratio larger than a threshold for a fixed model size, what will happen if we use different mixing ratio at different stages of training, will the model forget what it already memorize?

Could the authors provide results (for example Figure 2) with mixing ratio from 0 to 1 rather than from 0 to 0.4?

**Ethical Concerns:**

["NO or VERY MINOR ethics concerns only"]

**Limitations:**

Yes

**Quality:**

3

**Strengths And Weaknesses:**

Some interesting findings are revealed via the experiments in this paper. For instance, given a fixed data mixing, the model memorizes knowledge only when its model size is larger than threshold. Given a fixed model size, data mixing also has a threshold, below which leads to model memorizing nothing even with extensive training.

Some theoretical analysis are provided. The authors also discussed strategies to enhance knowledge acquisition under low mixing ratios.

---

> ### Author Rebuttal · Authors · 2025-07-31
>
> We sincerely thank the reviewer for the positive feedback and for recognizing the value of our findings. Below, we answer the reviewer's questions.
>
> ---
> **Q1:** What will happen if we use different mixing ratios at different stages of training? Will the model forget what it has already memorized?
>
> **A1:** Yes, the model can forget previously memorized knowledge if the mixing ratio of knowledge-dense data is reduced in later stages of training. The extent of forgetting depends on the new mixing ratio.
>
> Specifically, consider the following setup: we start from a model checkpoint $\mathcal{M}_1$, trained for $T_1$ tokens with a knowledge-dense mixing ratio $r_1$. We then continue training for an additional $T_2$ tokens using a smaller mixing ratio $r_2 < r_1$, resulting in a new checkpoint $\mathcal{M}_2$. The performance of $\mathcal{M}_2$ on the knowledge-dense data depends on $r_2$:
>
> - For moderate $r_2$, the model can retain the knowledge from $\mathcal{M}_1$ and keeps learning from the knowledge-dense data in the second stage, with $\mathcal{M}_2$ achieving higher accuracy than $\mathcal{M}_1$.
>
> - For small $r_2$, the model undergoes catastrophic forgetting, and the accuracy of $\mathcal{M}_2$ on knowledge-dense data drops below that of $\mathcal{M}_1$, even returning to near-zero levels when $r_2$ is too small.
>
>
> To verify this, we use a 70M Pythia model as $\mathcal{M}_1$, which has been trained for $T_1 = 24$B tokens on a mixture of FineWeb-Edu and SynBio-320k with $r_1 = 0.8$, achieving **62.6%** accuracy on SynBio. We then continue training with $r_2 < r_1$ for $T_2=32$B tokens. Results in Table 1 confirm the above arguments.
>
>
>
> **Table 1: Accuracy (%) on SynBio-320k with different mixing ratios during continual pre-training.**
> *CPT* stands for *Continual Pre-Training*. The initial checkpoint achieves 62.6% accuracy on SynBio.
>
> | Mixing Ratio ($r_2$) |  Acc. on SynBio(%) |
> |:----------------------:|:-------------------:|
> | 0.1                 | 0.4              |
> | 0.15                | 1.8               |
> | 0.2                 | 14.3               |
> | 0.25                | 35.3               |
> | 0.3                 | 50.0               |
> | 0.35                | 59.7               |
> | 0.4                 | 67.4               |
>
> ---
>
>
> **Q2:** Could the authors provide results (for example, Figure 2) with mixing ratio from 0 to 1 rather than from 0 to 0.4?
>
> **A2:** Sure! Following the setup in Figure 2(a), we conduct additional experiments for $r>0.4$ using 70M Pythia models. We summarize the results for $r$ from 0 to 1 in Table 2 below. Consistent with our findings in the paper, the accuracy on SynBio remains near zero for r<0.3 and quickly improves as r grows from 0.3 to 1. We will include the additional results for $r>0.4$ in our next version.
>
> **Table 2: Accuracy (%) on SynBio for 70M Pythia models trained on the mixture of FineWeb-Edu and SynBio-320k.**
>
> | Mixing Ratio $r$ | Acc. on SynBio (%) |
> |:----------------------:|:---------------:|
> | 0.1                 | 0.4           |
> | 0.15                 | 0.4          |
> | 0.2                 | 0.4          |
> | 0.25                 | 0.8          |
> | 0.3                  |8.7           |
> | 0.35                 |29.6          |
> | 0.4                 |40.1         |
> | 0.45                |  58.8        |
> | 0.5                 |   68.6     |
> | 0.6                 |   75.9   |
> | 0.7                 |     84.7     |
> | 0.8                 |     91.1 |
> | 0.9                 |       96.0   |
> | 1                   |       99.2     |
>
>
> ---
>
> We thank the reviewer again for the thoughtful and constructive feedback! We hope our responses have addressed all questions from the reviewer. If so, we kindly ask the reviewer to raise the rating. If the reviewer has any further questions, we will be more than happy to answer them.

---

> > ### Author Response · Authors · 2025-08-07
> >
> > Dear Reviewer,
> >
> > Thank you again for your time in reviewing our paper. For the two questions raised in your review (i.e., (1) how much knowledge can the model retain if we use different mixing ratios during different stages,  and (2) the model's performance for mixing ratios greater than 0.4), we have conducted additional experiments and included the results in our rebuttal. We believe we have adequately addressed your questions.
> >
> > As the author-reviewer discussion phase is approaching the deadline, we wanted to kindly check if you have any additional questions or concerns about our paper. We would be more than happy to respond to any further points you might raise.

---

### Decision · Program_Chairs · 2025-09-17

**Decision:**

Accept (spotlight)

**Comment:**

The paper studies how mixing between different types of data (knowledge-dense expert data vs web scrape) results for pretraining data results in transitions in scaling laws, and connect this transition to memorization. They also propose an information-theoretic model to predict these transitions.

Overall, the reviewers found the results novel and significant. The controlled experimental setup was highlighted as a strength, providing clear and well-presented results. Some minor issues around sensitivity to init, robustness of phase transition claims under a different loss, and clarity of definitions were addressed during the rebuttal phase.